# Linear Mixture Distributionally Robust Markov Decision Processes

**Zhishuai Liu**
Department of Biostatistics & Bioinformatics
Duke University
Durham, NC 27708
zhishuai.liu@duke.edu

**Pan Xu**
Department of Biostatistics & Bioinformatics
Duke University
Durham, NC 27708
pan.xu@duke.edu

## Abstract

Many real-world decision-making problems face the off-dynamics challenge: the agent learns a policy in a source domain and deploys it in a target domain with different state transitions. The distributionally robust Markov decision process (DRMDP) addresses this challenge by finding a robust policy that performs well under the worst-case environment within a pre-specified uncertainty set of transition dynamics. Its effectiveness heavily hinges on the proper design of these uncertainty sets, based on prior knowledge of the dynamics. In this work, we propose a novel linear mixture DRMDP framework, where the nominal dynamics is assumed to be a linear mixture model. In contrast with existing uncertainty sets directly defined as a ball centered around the nominal kernel, linear mixture DRMDPs define the uncertainty sets based on a ball around the mixture weighting parameter. We show that this new framework provides a more refined representation of uncertainties compared to conventional models based on $(s, a)$-rectangularity and $d$-rectangularity, when prior knowledge about the mixture model is present. We propose a meta algorithm for robust policy learning in linear mixture DRMDPs with general $f$-divergence defined uncertainty sets, and analyze its sample complexities under three divergence metrics instantiations: total variation, Kullback-Leibler, and $\chi^2$ divergences. These results establish the statistical learnability of linear mixture DRMDPs, laying the theoretical foundation for future research on this new setting.

## 1 Introduction

In off-dynamics reinforcement learning (RL) [21, 48, 10, 45], an agent trains a policy in a source domain and then deploys the learned policy in an unknown target domain with possible dynamics shift from the source one. The dynamics shift may arise from the sim-to-real gap [34, 57, 7] when the policy is trained on a simulator that mimics the real environment. The dynamics shift may cause catastrophic failure since the policies learned from the source domain would admit distinct behaviors in the target domain which thereby deviate the agent from achieving its goal, e.g., maximizing a long term cumulative reward in the target domain.

A line of literature have tried to solve off-dynamics RL problems from various perspectives differentiated by the extend of information we can collect from the source and target domains [21, 48, 10, 45, 24]. In this work, we are particularly interested in the case where we can only collect data from the source domain for policy learning. In other words, the agent has little knowledge about the target domain but only holds the belief that its dynamics should be structurally close to the source domain. Such large uncertainty in the target domain necessitates robust policy learning to train policies that generalize well across a wide range of target domain. To this end, the agent usually has to adopt a pessimism philosophy in the face of large uncertainty to prepare for the worst case dynamics.

39th Conference on Neural Information Processing Systems (NeurIPS 2025).

In this work, we tackle off-dynamics RL through a framework widely studied in recent decades that adopts the pessimism philosophy described above, the distributionally robust Markov decision process (DRMDP) [38, 18, 32, 49, 47]. In DRMDPs, the term dynamics specifically refers to the transition kernel $P$ that takes as input the state-action pair and outputs the next state. The uncertainty of the target domain dynamics is modeled as an *uncertainty set* defined by some probability measure centered around the transition kernel $P^0$ of the source domain. Given a specific construction of the uncertainty set, an agent aims to learn a robust policy that secures a fair amount of cumulative reward uniformly across all target domains represented within the uncertainty set. In essence, the DRMDP formulates a max-min optimization problem, under which the goal is to find a policy that is guaranteed to perform well even in the worst-case environment within the uncertainty set. Recent studies [18, 47, 12] have shown that both the tractability of the max-min optimization problem and the performance of the learned robust policy depend heavily on the design of the uncertainty set. On the one hand, Wiesemann et al. [47] show that arbitrary uncertainty sets could render the max-min optimization problem NP-hard. On the other hand, an appropriately designed uncertainty set—based on suitable prior information—can effectively model distributional shifts and prevent the robust policy from becoming overly conservative.

To demonstrate this further, let us first consider the case where no prior information on the structure of transitions is available. It is then reasonable to independently construct uncertainty sets centered around transitions at each state-action pair, i.e., $\mathcal{U}(P^0) = \bigotimes_{(s,a)} \mathcal{U}(P^0(\cdot|s,a))$. This gives us the commonly studied $(s,a)$-rectangular uncertainty set [18, 52, 16]. It is well-known that $(s,a)$-rectangular uncertainty sets may include transitions that would never occur in the target domain—particularly when the state-action space is large, as is often the case in many real-world applications—resulting in overly conservative policies [12, 29, 25]. To address this issue, one option is to incorporate transition structure information into the uncertainty set design.

A commonly studied class of transition is the mixture distribution [19, 2, 6, 58, 22], which frequently arises in practice [30, 35, 36]. Assuming we are equipped with the prior information that the source domain transition kernels are linear mixture distributions of some basis modes $\phi(\cdot|s,a)$ that we have access to, i.e., $P^0(\cdot|s,a) = \langle \boldsymbol{\theta}^0, \phi(\cdot|s,a) \rangle$, where $\boldsymbol{\theta}^0$ is some unknown mixture weighting parameter. Then it is reasonable to hold the belief that the target domain dynamics maintains the linear mixture structure, i.e., $P(\cdot|s,a) = \langle \boldsymbol{\theta}, \phi(\cdot|s,a) \rangle$, while the parameter $\boldsymbol{\theta}$ is subject to some perturbation from $\boldsymbol{\theta}^0$. This kind of perturbation cannot be precisely characterized by existing uncertainty set designs in literature. In this work, we propose the novel *linear mixture uncertainty set*. We formally establish a new framework for DRMDPs with linear mixture uncertainty sets, dubbed as the *linear mixture DRMDP*. This formulation is intrinsically different from existing frameworks due to the introduction of structural information into both the dynamics and the uncertainty set design. Focusing on the offline RL setting where we only have access to an offline dataset pre-collected from the source domain by a behavior policy $\pi^b$, we provide answers to the following fundamental questions:

***How many samples are required to learn an $\epsilon$-optimal robust policy for linear mixture DRMDPs?***

In this paper, we provide the first ever study on linear mixture distributionally robust Markov decision processes. We summarize our main contributions as follows:

- We show that the novel design of the linear mixture uncertainty set can achieve more refined quantification of the dynamics shift compared to the standard $(s,a)$-rectangular and the $d$-rectangular uncertainty set, hence could potentially be more favorable. This justifies the need of linear mixture DRMDPs. Further, we prove that the dynamic programming principles hold for linear mixture DRMDPs, which motivate the algorithm design and theoretical analysis.
- We propose a meta algorithm based on the double pessimism principle [5] and transition targeted ridge regression [22] for linear mixture DRMDPs with general probability divergence metric defined uncertainty sets. From the theoretical side, we prove that when instantiating to the commonly studied TV, KL and $\chi^2$ divergences, our proposed algorithm achieves upper bound on the suboptimality in order of $\tilde{O}(dH^2 C^{\pi^\star}/\sqrt{K})$, $\tilde{O}(dH^2 C^{\pi^\star} e^{H/\underline{\lambda}}/\rho\sqrt{K})$ and $\tilde{O}(d(\sqrt{\rho}H^3 + H^2)C^{\pi^\star}/\sqrt{K})$[1] respectively, showing the statistical learnability of linear mixture DRMDPs.

---

[1]Here $d$ is the number of basis modes, $H$ is the horizon length, $C^{\pi^\star}$ is a coverage parameter of the offline dataset (see Assumption 5.1), $\underline{\lambda}$ is the lower bound on the dual variable of KL-divergence (see Assumption 5.5), $\rho$ is the uncertainty level and $K$ is the number of samples in the offline dataset.

**Notations** We provide the notations used in this paper for the reader's reference. We denote $\Delta(\mathcal{S})$ as the set of probability measures over some set $\mathcal{S}$. For any number $H \in \mathbb{Z}_+$, we denote $[H]$ as the set of $\{1, 2, \cdots, H\}$. For any function $V : \mathcal{S} \to \mathbb{R}$, we denote $[\mathbb{P}_h V](s, a) = \mathbb{E}_{s' \sim P_h(\cdot|s,a)}[V(s')]$ as the expectation of $V$ with respect to the transition kernel $P_h$, and $[V(s)]_\alpha = \min\{V(s), \alpha\}$, given a scalar $\alpha > 0$, as the truncated value of $V$. For a vector $\boldsymbol{x}$, we denote $x_j$ as its $j$-th entry. And we denote $[x_i]_{i \in [d]}$ as a vector with the $i$-th entry being $x_i$. For a matrix $A$, denote $\lambda_i(A)$ as the $i$-th eigenvalue of $A$. For two matrices $A$ and $B$, we denote $A \preceq B$ as the fact that $B - A$ is a positive semidefinite matrix, and $A \succeq B$ as the fact that $A - B$ is a positive semidefinite matrix. For any function $f : \mathcal{S} \to \mathbb{R}$, we denote $\|f\|_\infty = \sup_{s \in \mathcal{S}} f(s)$. Denote $\Delta^{d-1}$ as the $d-1$ dimensional simplex. For any two probability distributions $P$ and $Q$ on $\mathcal{S}$ such that $P$ is absolutely continuous with respect to $Q$, we define the total variation (TV) divergence as $D_{\text{TV}}(P||Q) = 1/2 \int_{\mathcal{S}} |P(s) - Q(s)| ds$, the Kullback-Leibler divergence as $D_{\text{KL}}(P||Q) = \int_{\mathcal{S}} P(s) \log P(s)/Q(s) ds$ and the $\chi^2$ divergence as $D_{\chi^2}(P||Q) = \int_{\mathcal{S}} (P(s) - Q(s))^2/Q(s) ds$.

## 2   Related works

A substantial body of empirical research also explores off-dynamics RL through the lens of domain adaptation and transfer learning [10, 8, 54, 50, 46, 14, 45, 28, 7, 13], among others. In this paper, we focus on the DRMDP formulation of off-dynamics RL. Readers are referred to the aforementioned works for complementary approaches that explore this orthogonal line of research.

Several lines of studies have extensively studied DRMDPs from different perspectives [49, 61, 3, 53, 5, 41, 27, 25, 42, 26], including offline robust policy learning, online data exploration, and function approximation, etc. We particularly focus on works studying uncertainty set design in this part. The seminal works of [18, 33] proposed the DRMDP with $(s, a)$-rectangularity, where the uncertainty set is constructed independently at each state-action pair. Wiesemann et al. [47] then studied the $s$-rectangular uncertainty set, which includes the $(s, a)$-rectangular uncertainty set as a special case. They also showed that solving DRMDPs with general uncertainty sets can be NP-hard. When the state and action spaces are large, DRMDPs with $s$-rectangular and $(s, a)$-rectangular uncertainty sets suffer from issues of conservative policies and intractable computation complexity. Goyal and Grand-Clement [12] proposed to leverage the latent structure of the transition kernel and propose the $r$-rectangular uncertainty set, which was then shown to be significantly less conservative than prior approaches. Motivated by the design of $r$-rectangular uncertainty set, Ma et al. [29] proposed the setting of $d$-rectangular linear DRMDPs, where the transition kernel is assumed to be a linear combination of a known feature mapping and unknown factor distributions, and the uncertainty is constructed upon perturbations onto the factor distributions.

Beyond the conventional rectangularity framework, Zhou et al. [60] proposed two novel uncertainty set, one is based on the double sampling and the other on an integral probability metric, to achieve more efficient and less conservative robust policy learning. Zouitine et al. [62] proposed the time-constraint DRMDPs, where the uncertainty set is modeled to be time-dependent, to accurately reflect real-world dynamics and thus solve the conservativeness issue. Li et al. [23] studied DRMDPs with non-rectangular uncertainty sets, where dynamic programming principles do not hold, and they provided a policy gradient algorithm to learn the optimal robust policy. Our work also aims to address the issues of conservativeness and computational tractability in DRMDPs. However, we tackle this problem from a completely different perspective. Specifically, with appropriate prior information, we assume that the uncertainty originates from perturbations in the underlying parameters that define the model. Given the great interest in linear mixture model from both practical [20, 11] and theoretical [2, 19, 58] sides, our linear mixture DRMDP framework provides the first result on robust policies learning when the source and target transitions are linear mixture model.

## 3   Linear mixture distributionally robust Markov decision process

An episodic distributionally robust MDP is denoted as DRMDP$(\mathcal{S}, \mathcal{A}, H, \mathcal{U}^\rho(P^0), r)$, with the horizon length $H$, time homogeneous nominal transition kernel $P^0 = \{P_h^0\}_{h=1}^H$, deterministic reward function $r = \{r_h\}_{h=1}^H$ and uncertainty set $\mathcal{U}^\rho(P^0) = \otimes_{h \in [H]} \mathcal{U}_h^\rho(P_h^0)$. The robust value function and

robust Q-function are defined as

$$V_{h,P^0}^{\pi,\rho}(s) = \inf_{P \in \mathcal{U}(P^0)} E^P\left[\sum_{t=h}^{H} r_t(s_t, a_t) \big| s_h = s, \pi\right], \tag{3.1}$$

$$Q_{h,P^0}^{\pi,\rho}(s,a) = \inf_{P \in \mathcal{U}(P^0)} E^P\left[\sum_{t=h}^{H} r_t(s_t, a_t) \big| s_h = s, a_h = a, \pi\right].$$

We assume transition kernels are mixture distributions in the following assumption.

**Assumption 3.1** (Linear Mixture Models [2]). *For any $(s, a, h) \in \mathcal{S} \times \mathcal{A}$, there exists a feature mapping $\phi : \mathcal{S} \times \mathcal{A} \to (\Delta(\mathcal{S}))^d$, where $\phi(\cdot|s,a) = \left[\phi_1(\cdot|s,a), \cdots, \phi_d(\cdot|s,a)\right]^\top$ and $\phi_i(\cdot|s,a) \in \Delta(\mathcal{S}), \forall i \in [d]$. Assume there exists a $d$-dimensional vector $\boldsymbol{\theta}_h^0$, where $\boldsymbol{\theta}_h^0 \in \Delta^{d-1}$, such that*

$$P_h^0(\cdot|s,a) = \left\langle \boldsymbol{\phi}(\cdot|s,a), \boldsymbol{\theta}_h^0 \right\rangle, \quad \forall (s,a) \in \mathcal{S} \times \mathcal{A}. \tag{3.2}$$

$\phi_i(\cdot|s,a)$ is often referred to as the basis latent mode [19, 2]. The nominal transition $P_h^0(\cdot|s,a)$ is a probabilistic mixture of the basis latent modes, and $\boldsymbol{\theta}_h^0$ are the weighting parameters. We assume the agent knows the basis latent modes, but doesn't know $\boldsymbol{\theta}_h^0$. In the literature, there are several kinds of assumptions on parameter norms. The current formulation leads to $\|\boldsymbol{\theta}_h^0\|_2 \le 1$, and $\|\phi_i(\cdot|s,a)\|_1 = 1$, for any $(i, s, a) \in [d] \times \mathcal{S} \times \mathcal{A}$. For any function $V : \mathcal{S} \to [0, 1]$ and $(s, a) \in \mathcal{S} \times \mathcal{A}$, we denote $\phi^V(s,a) = \int_{\mathcal{S}} \phi(s'|s,a) V(s') ds'$, then we have $\|\phi^V(s,a)\|_2 \le \sqrt{d}$.

Now given the structure of the nominal kernel $P^0$, we define the uncertainty set. In particular, we assume the parameter $\boldsymbol{\theta}$ can be perturbed in the test environment, that is, $\mathcal{U}_h^\rho(P^0) = \otimes_{(s,a) \in \mathcal{S} \times \mathcal{A}} \mathcal{U}^\rho(s, a; \boldsymbol{\theta}_h^0)$, where

$$\mathcal{U}^\rho(s, a; \boldsymbol{\theta}_h^0) = \left\{ P(\cdot|s,a) \in \Delta(\mathcal{S}) \big| P(\cdot|s,a) = \left\langle \boldsymbol{\phi}(\cdot|s,a), \boldsymbol{\theta} \right\rangle : \boldsymbol{\theta} \in \Delta^{d-1}, D(\boldsymbol{\theta}||\boldsymbol{\theta}_h^0) \le \rho \right\}, \tag{3.3}$$

$D(\cdot||\cdot)$ is a probability divergence metric that will be instantiated later. Denote $\boldsymbol{\Theta}_h = \{\boldsymbol{\theta}_h \in \Delta^{d-1} | D(\boldsymbol{\theta}||\boldsymbol{\theta}_h^0) \le \rho\}$ as the parameter uncertainty set. Notably, the uncertainty set of the transition kernel, $\mathcal{U}_h^\rho(P^0)$, is determined by the uncertainty set of the weighting parameter, $\boldsymbol{\Theta}_h$. We refer to the uncertainty set defined in (3.3) as the *linear mixture uncertainty set* and the DRMDP equipped with the linear mixture uncertainty set as the linear mixture DRMDP. We highlight due to the fact that the perturbation on parameters $\boldsymbol{\theta}_h$ are decoupled among times steps, the linear mixture uncertainty set belongs to the family of rectangular-type uncertainty sets, which also includes the $(s, a)$-rectangularity [18], $s$-rectangularity [47] and $r$-rectangularity [12], etc. It is well known that the rectangularity property ensures the dynamic programming principles hold for DRMDPs. Following the proof of Propositions 3.2 and 3.3 in [24], we establish the (optimal) robust Bellman equation and the existence of deterministic optimal policy.

**Proposition 3.2** (Robust Bellman Equation). *Under the linear mixture DRMDP setting, for any nominal transition kernel $P^0$ and any stationary policy $\pi = \{\pi_h\}_{h=1}^H$, the following robust Bellman equation holds: for any $(h, s, a) \in [H] \times \mathcal{S} \times \mathcal{A}$,*

$$Q_h^{\pi,\rho}(s,a) = r_h(s,a) + \inf_{P_{h,s,a} \in \mathcal{U}_h^\rho(s,a;\boldsymbol{\theta}_h^0)} \mathbb{E}_{s' \in P_{h,s,a}}\left[V_{h+1}^{\pi,\rho}(s')\right],$$

$$V_h^{\pi,\rho}(s) = \mathbb{E}_{a \sim \pi_h(\cdot|s)}\left[Q_h^{\pi,\rho}(s,a)\right].$$

**Proposition 3.3** (Existence of the optimal policy). *Assume the nominal transition kernel $P^0$ satisfies (3.2) and the uncertainty set satisfies (3.3). Then there exists a deterministic and stationary pilicy $\pi^\star$ such that $V_h^{\pi^\star,\rho}(s) = V_h^{\star,\rho}(s)$ and $Q_h^{\pi^\star,\rho}(s,a) = Q_h^{\star,\rho}(s,a)$, for any $(h, s, a) \in [H] \times \mathcal{S} \times \mathcal{A}$.*

Then we have the robust Bellman optimality equation:

$$Q_h^{\pi,\rho}(s,a) = r_h(s,a) + \inf_{P_h(\cdot|s,a) \in \mathcal{U}_h^\rho(s,a;\boldsymbol{\theta}_h^0)} \mathbb{E}_{s' \in P_h(\cdot|s,a)}\left[V_{h+1}^{\star,\rho}(s')\right],$$

$$V_h^{\pi,\rho}(s) = \max_{a \in \mathcal{A}} Q_h^{\pi,\rho}(s,a).$$

Then it suffices to estimate the optimal robust Q-function $Q_h^{\star,\rho}$ to find $\pi^\star$.

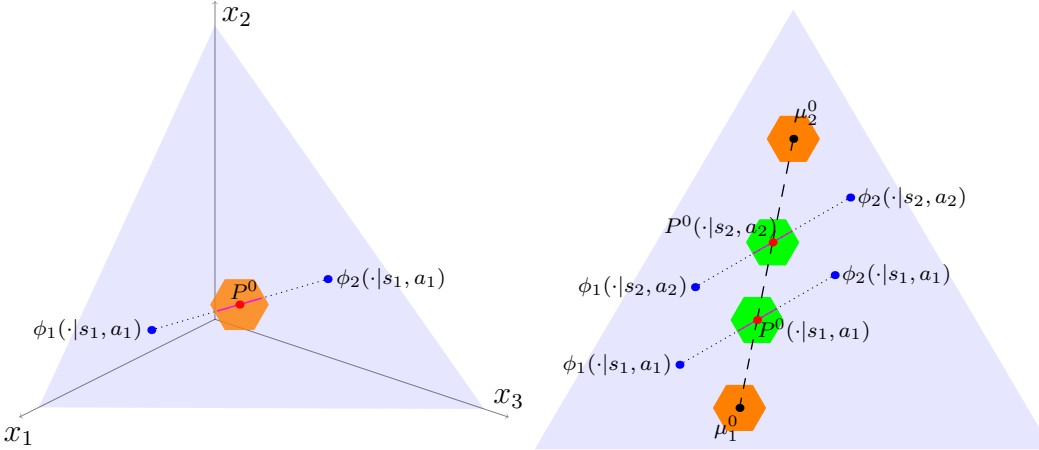

(a) Comparison with $(s,a)$-rectangular uncertainty set.     (b) Comparison with $d$-rectangular uncertainty set.

Figure 1: Illustrations of the comparison among $(s,a)$-rectangular, $d$-rectangular, and linear mixture uncertainty sets in $\mathbb{R}^3$, where $\mathcal{S} = \{x_1, x_2, x_3\}$. The light blue region represents the probability simplex and each point in the region is a probability distribution. $\phi_1$ and $\phi_2$ are two basis modes, which are represented by blue dots. The nominal kernel $P^0$ is represented by red dots, with $\boldsymbol{\theta} = [1/2, 1/2]^\top$. The linear mixture uncertainty set with radius $\rho = 1/8$ is the magenta segment. (a) The orange hexagons ⬣ represents the smallest $(s,a)$-rectangular uncertainty set centered around $P^0$ that cover the linear mixture uncertainty set. (b) The orange hexagons ⬣ represent the uncertainty sets centered around factor distributions $\mu_1^0$ and $\mu_2^0$. The green hexagons ⬣ represent the $d$-rectangular uncertainty sets, which are linear combinations of the uncertainty sets of the factor distributions and cover the linear mixture uncertainty set.

**Offline dataset and learning goal**   Denote $\mathcal{D} = \{(s_h^k, a_h^k, s_{h+1}^k)\}_{h,k=1}^{H,K}$ as the offline dataset consisting $K$ trajectories collected from the nominal environment by the behavior policy $\pi^b$. The goal of the agent is to learn an optimal robust policy $\pi^{\star,\rho}$ only using the offline dataset $\mathcal{D}$. Denote the learned policy as $\hat{\pi}$, we define the suboptimality of $\hat{\pi}$ as $\text{SubOpt}(\hat{\pi}, s_1, \rho) := V_1^{\star,\rho}(s_1) - V_1^{\hat{\pi},\rho}(s_1)$, which is used to evaluate the performance of the estimated policy $\hat{\pi}$.

### 3.1   Comparison with $(s,a)$-rectangular uncertainty sets

Now we compare our linear mixture uncertainty set with the most commonly studied $(s,a)$-rectangular uncertainty set [18, 32]. For the ease of illustration, we focus on the total variation (TV) divergence.

**Lemma 3.4.** *When the probability is specified to the TV divergence for both the linear mixture uncertainty set and the $(s,a)$-rectangular uncertainty set, the regular $(s,a)$-rectangular uncertainty set can be strictly more conservative.*

As illustrated by the example in Figure 1a, the TV divergence defined linear mixture uncertainty set is strictly smaller than the TV divergence defined $(s,a)$-rectangular uncertainty set. We conclude that (1) the linear mixture uncertainty set can achieve refined representations of the dynamics perturbation, and (2) the size and shape of the linear mixture uncertainty set essentially depends on the pre-specified basis latent modes. Finally, we show in the following lemma that the linear mixture uncertainty set can be adapted to recover the standard $(s,a)$-rectangular uncertainty set.

**Lemma 3.5.** *In tabular DRMDPs with finite state and action spaces, the linear mixture uncertainty set can recover the standard $(s,a)$-rectangular uncertainty set by design.*

### 3.2   Comparison with $d$-rectangular uncertainty sets

Another well-studied uncertainty set with linear structure is the $d$-rectangular uncertainty set [12, 29, 24, 25, 44]. Next we explore the relation between the $d$-rectangular uncertainty set and the linear mixture uncertainty set. Given the fact that the standard linear MDP and standard linear mixture

MDP do not cover each other [59], it is not straightforward to compare the more complex uncertainty sets. However, we find a case that the $d$-rectangular uncertainty set can be more conservative, as illustrated in Figure 1b. Specifically, $\mathcal{S} = \{s_1, s_2, s_3\}$, $\boldsymbol{\mu}^0 = [\mu_1^0, \mu_2^0]^\top$ is the factor distribution and the orange hexagons are factor uncertainty sets [24] of the $d$-rectangular linear DRMDP. There exists a feature mapping $\boldsymbol{\psi}(\cdot, \cdot) : \mathcal{S} \times \mathcal{A} \to \mathbb{R}^2$ such that $P^0(\cdot|s, a) = \langle \boldsymbol{\psi}(s, a), \boldsymbol{\mu}^0(\cdot) \rangle$. Moreover, let $\boldsymbol{\theta}^0 = [1/2, 1/2]^\top$ and $\boldsymbol{\phi}(\cdot|\cdot, \cdot)$ be basis modes such that $P^0(\cdot|s, a) = \langle \boldsymbol{\phi}(\cdot|s, a), \boldsymbol{\theta}^0 \rangle$ in the linear mixture DRMDP. We can observe that the smallest $d$-rectangular uncertainty sets covering the linear mixture uncertainty sets are the green hexagons. We can conclude that the $d$-rectangular uncertainty set is strictly larger than the linear mixture uncertainty set in this instance.

**Remark 3.6.** *We note that the comparison with $(s, a)$-rectangular uncertainty sets and $d$-rectangular uncertainty sets is only to show that in particular cases, the linear mixture uncertainty set has advantages in modeling the uncertainty while existing uncertainty sets fall short. It does not lead to the conclusion linear mixture is always better or always less conservative. We highlight the linear mixture DRMDP assumes prior information, i.e., the linear mixture dynamics and known basis modes, it's not fair to directly compare it with other DRMDP settings because they either require other prior information, e.g., the linear MDP structure and feature mapping $\boldsymbol{\psi}$ for $d$-rectangular DRMDP, or do not require prior information at all, e.g., the $(s, a)$-rectangular DRMDP.*

## 4 Algorithm design

In this section, we design a meta algorithm to learn the optimal robust policies under various probability divergence metrics. Due to technical reasons (see Remark 4.2), we assume that for each state $s \in \mathcal{S}$, there are finitely reachable states.

**Assumption 4.1.** $\forall (h, s, a) \in [H] \times \mathcal{S} \times \mathcal{A}$, *we denote the feasible set corresponding to $(s, a)$ as* $\mathcal{S}_h(s, a) := \{s' \in \mathcal{S} | P_h^0(s'|s, a) > 0\}$. *Assume there exists a positive constant $p_{\min} > 0$, such that* $\min_{s' \in \mathcal{S}_h(s,a)} P_h^0(s'|s, a) > p_{\min}, \forall (h, s, a) \in [H] \times \mathcal{S} \times \mathcal{A}$.

A direct implication of Assumption 4.1 is that for any $(h, s, a) \in [H] \times \mathcal{S} \times \mathcal{A}$, we can construct a feasible set $S_h(s, a)$ with cardinality $|S_h(s, a)| = \lceil 1/p_{\min} \rceil$ that contains $\mathcal{S}_h(s, a)$, i.e., $\mathcal{S}_h(s, a) \subset S_h(s, a)$. Specifically, for $s' \in S_h(s, a)/\mathcal{S}_h(s, a)$, we have $P_h(s'|s, a) = 0$. Let $\boldsymbol{\delta}_s \in \{0, 1\}^{\lceil 1/p_{\min} \rceil}$ be the one-hot vector with $\boldsymbol{\delta}_s(s) = 1$, we propose to estimate the mixture weights $\{\boldsymbol{\theta}_h^0\}_{h=1}^H$ by solving the regularized transition-targeted regression:

$$\min_{\boldsymbol{\theta} \in \mathbb{R}^d} \sum_{k=1}^K \sum_{s \in S_h(s_h^k, a_h^k)} \left( \boldsymbol{\phi}(s|s_h^k, a_h^k)^\top \boldsymbol{\theta} - \boldsymbol{\delta}_{s_{h+1}^k}(s) \right)^2 + \lambda \|\boldsymbol{\theta}\|_2^2. \tag{4.1}$$

Then we have the close form solution $\hat{\boldsymbol{\theta}}_h^0 = \boldsymbol{\Lambda}_h^{-1} \boldsymbol{b}_h$, where

$$\boldsymbol{\Lambda}_h = \sum_{k=1}^K \sum_{s \in S_h(s_h^k, a_h^k)} \boldsymbol{\phi}(s|s_h^k, a_h^k) \boldsymbol{\phi}(s|s_h^k, a_h^k)^\top + \lambda \mathbf{I}_d, \boldsymbol{b}_h = \sum_{k=1}^K \sum_{s \in S_h(s_h^k, a_h^k)} \boldsymbol{\delta}_{s_{h+1}^k}(s) \boldsymbol{\phi}(s|s_h^k, a_h^k).$$

**Remark 4.2.** *Though the most commonly studied method for parameter estimation in standard linear mixture MDP literature is the value-targeted regression [2, 58, 55], we find a data coverage issue that is hard to bypass in the suboptimality analysis for algorithms using the value-targeted regression for parameter estimation. Instead, we propose to substitute the value-target by the transition information [56, 22]. With Assumption 4.1, we are able to construct feasible sets $S_h(s, a)$ and conduct the transition-targeted ridge regression (4.1). Besides the difference in the target, the transition-targeted regression estimation also induces a notable problem in the concentration analysis. To see this, we note that typically we resort to the self-normalized concentration lemma for vector-valued martingales [1, Theorem 1] to bound the estimation error of the value-targeted regression. However, as discussed in [22], the errors, $\boldsymbol{\epsilon}_h^k = [P(s|s_h^k, a_h^k) - \delta_{s_{h+1}^k}(s)]_{s \in \mathcal{S}_h(s_h^k, a_h^k)}, \forall k \in [K]$, in the transition-targeted ridge regression are not independent due to the fact that $\sum_{s \in \mathcal{S}_h(s_h^k, a_h^k)} \boldsymbol{\epsilon}_h^k(s) = 0$. Thus, the self-normalized concentration lemma in [1] does not apply anymore, as the independence between errors is an essential condition. To solve this issue, we resort to the concentration lemma proposed by [22], which is specifically tailored to the dependent error structure in (4.1). With the concentration result, we can construct confidence sets as follows.*

Define the confidence set $\boldsymbol{\Theta}_h = \{\boldsymbol{\theta} \in \mathbb{R}^d | \|\boldsymbol{\theta} - \hat{\boldsymbol{\theta}}_h^0\|_{\boldsymbol{\Lambda}_h} \leq \beta_h\}$, where the radius $\beta_h$ is to be determined. By adapting the Lemma 2 of [22] (see Lemma D.2 for details), we immediately have the following lemma stating that, with high probability, the true parameter lies in $\boldsymbol{\Theta}_h$.

**Lemma 4.3.** *Let $\zeta \in (0,1)$. For all $h \in [H]$, if we set $\beta_h = \frac{5}{4}\sqrt{\lambda} + \frac{2}{\sqrt{\lambda}}(2\log\frac{H}{\zeta} + d\log(4 + 4\lceil 1/p_{\min}\rceil K/\lambda d))$, then with probability at least $1 - \zeta$, it holds that $\boldsymbol{\theta}_h^0 \in \boldsymbol{\Theta}_h$.*

Given the confidence set of the parameter, next we define the confidence region for transition kernel. Specifically, denote

$$\widehat{\mathcal{P}} = \otimes_{h \in [H]} \widehat{\mathcal{P}}_h \text{ and } \widehat{\mathcal{P}}_h = \{\phi(\cdot|\cdot,\cdot)^\top \boldsymbol{\theta}_h | \boldsymbol{\theta}_h \in \widehat{\boldsymbol{\Theta}}_h\} \tag{4.2}$$

as the confidence region, where $\widehat{\boldsymbol{\Theta}}_h = \boldsymbol{\Theta}_h \cap \Delta^{d-1}$. Leveraging the double pessimism principle proposed in [5], we define the value estimator as

$$J_{\text{Pess}^2}(\pi) := \inf_{P_h \in \widehat{\mathcal{P}}_h, 1 \leq h \leq H} \inf_{\tilde{P}_h \in \mathcal{U}_h^\rho(P_h)} V_{1,\tilde{P}}^\pi(s_1),$$

where $V_{1,\tilde{P}}^\pi(s_1)$ represents the robust value function, with the nominal kernel $\tilde{P}$. The policy that maximizes the doubly pessimistic value estimator as the estimated optimal robust policy,

$$\hat{\pi} := \underset{\pi \in \Pi}{\arg\max} \, J_{\text{Pess}^2}(\pi). \tag{4.3}$$

We present our meta-algorithm in Algorithm 1.

---

**Algorithm 1** Meta Algorithm of Policy Optimization for Linear Mixture DRMDP

---

1: **Input:** The offline dataset $\mathcal{D}$, the regularizer $\lambda$, and the robust level $\rho$.
2: Construct the confidence region $\widehat{\mathcal{P}}$ according to (4.2).
3: Get the estimated optimal robust policy $\hat{\pi}$ by (4.3).
4: **Return:** Policy $\hat{\pi}$.

---

## 5 Theoretical guarantees

In this section, we provide corresponding finite sample guarantees for Algorithm 1. In particular, we instantiate the probability divergence metric $D(\cdot|\cdot)$ in (3.3) as the TV, KL and $\chi^2$-divergences, which are most commonly studied in literature [18, 52, 42].

### 5.1 TV-divergence

Before we present the result on the finite sample suboptimality upper bound, we introduce the following coverage assumption on the offline dataset.

**Assumption 5.1** (Robust Partial Coverage: TV-divergence). *For any $P \in \widehat{\mathcal{P}}$, denote the worst-case transition at $(s,a)$ as*

$$P_h^{\pi^\star,\dagger}(\cdot|s,a) = \underset{\tilde{P}_h \in \mathcal{U}^\rho(P_h^0)}{\arg\inf} \mathbb{E}_{s' \sim \tilde{P}_h(\cdot|s,a)}[V_{h+1,P}^{\pi^\star,\rho}(s')], \tag{5.1}$$

*the problem dependent robust value covariance matrix as*

$$\boldsymbol{\Lambda}_h^{TV}\big(\alpha; d_{h,P^{\pi^\star,\dagger}}^{\pi^\star}, V_{h+1,P}^{\pi^\star}\big) = \mathbb{E}_{(s_h,a_h) \sim d_{h,P^{\pi^\star,\dagger}}^{\pi^\star}}\left[\big[\phi^{V_{h+1,P}^{\pi^\star}}(s_h,a_h)\big]_\alpha \big[\phi^{V_{h+1,P}^{\pi^\star}}(s_h,a_h)\big]_\alpha^\top\right], \tag{5.2}$$

*and the sampling covariance matrix as*

$$\boldsymbol{\Lambda}_h^0 = \sum_{k=1}^K \phi(s_{h+1}^k|s_h^k,a_h^k)\phi(s_{h+1}^k|s_h^k,a_h^k)^\top. \tag{5.3}$$

*Then we assume there exists a positive constant $C^{\pi^\star} > 0$, such that*

$$\sup_{\alpha \in [0,H]} \sup_{P \in \widehat{\mathcal{P}}} \sup_{x \in \mathbb{R}^d} \frac{x^\top \boldsymbol{\Lambda}_h^{TV}\big(\alpha; d_{h,P^{\pi^\star,\dagger}}^{\pi^\star}, V_{h+1,P}^{\pi^\star}\big)x}{x^\top \boldsymbol{\Lambda}_h^0 x} \leq \frac{C^{\pi^\star} H^2}{K}.$$

**Remark 5.2.** *Assumption 5.1 is a robust partial type coverage assumption on the offline dataset. It only requires the offline dataset have good coverage on the state-action space visited by the optimal robust policy $\pi^\star$, and it considers the worst case transition in (5.1). Assumption 5.1 resembles the partial coverage assumption for standard linear mixture MDP in [43] (see Table 1 in their paper for more details). Nevertheless, we highlight two major differences arising from the robust setting we considered in this work. First, the value covariance matrix $\mathbf{\Lambda}_h^{TV}$ is defined under the worst case transition. Second, there is an additional supremum over $\alpha$ which arises from the dual formulation of the TV divergence defined robust value function. Lastly, Assumption 5.1 considers all transitions in the confidence region $\widehat{\mathcal{P}}$, which depends on the offline dataset. Thus, it implicitly imposes a constraint on the offline dataset. We note that $\widehat{\mathcal{P}}$ can be replaced by the set of all feasible transition kernels $\mathcal{M} = \{\phi(\cdot|s,a)^\top \boldsymbol{\theta} : \boldsymbol{\theta} \in \mathbb{R}^d, \sum_{i=1}^d \theta_i = 1\}$, which would lead to a stronger assumption though.*

**Theorem 5.3** (TV-divergence). *Assume Assumptions 3.1, 4.1 and 5.1 hold. For $\zeta \in (0,1)$, there exists an absolute constant $c > 0$, such that if we set $\lambda = d$ in Algorithm 1, then for TV-divergence uncertainty set, with probability at least $1 - \zeta$, we have*

$$SubOpt(\hat{\pi}, s_1) \leq cdH^2 C^{\pi^\star} K^{-1/2} \log(K/p_{\min}d^2\zeta).$$

This result is the first of its kind since we are studying a new framework. Nevertheless, we can compare it with results for standard linear mixture MDPs. We note that [2] present a suboptimality bound in the order of $\tilde{O}(dH^{3/2}\sqrt{K})$. They assume in their work the transition is homogeneous, say, $P_1 = \cdots = P_H = P$, which means the weighting parameter $\boldsymbol{\theta}$ is shared across stages. This would reduce their bound by $\sqrt{H}$, which is indicated by [58]. If their analysis is modified to inhomogeneous transitions, an additional $O(\sqrt{H})$ factor would be added up. Thus, the bound in Theorem 5.3 matches that in Theorem 1 of [2] in terms of dimension $d$ and horizon length $H$. Next, we translate the suboptimality bound in Theorem 5.3 to the sample complexity bound.

**Corollary 5.4.** *Under the same assumptions and setting as in Theorem 5.3, to learn an $\epsilon$-optimal policy with probability at least $1 - \zeta$, we require the sample size $K$ satisfying*

$$K = \tilde{O}\Big(\frac{d^2 H^4 (C^{\pi^\star})^2}{\epsilon^2}\Big).$$

## 5.2 KL-divergence

For KL divergence, we require different assumptions due to its distinct geometry and dual formulation.

**Assumption 5.5** (Regularity of the KL-divergence duality variable). *We assume that the optimal dual variable $\lambda^\star$ for*

$$\sup_{\lambda \in \mathbb{R}_+} \big\{ - \lambda \log \big(\mathbb{E}_{i \sim \boldsymbol{\theta}_h}\big[ \exp \big\{ - \phi_i^{V_{h+1,P}^{\pi^\star}}/\lambda \big\}\big]\big) - \lambda\rho \big\}$$

*is lower bounded by $\underline{\lambda} > 0$ for any $\boldsymbol{\theta}_h \in \Delta^{d-1}$, $P = \{\phi(\cdot|\cdot,\cdot)^\top \boldsymbol{\theta}_h\}_{h=1}^H \in \mathcal{P}_\mathcal{M}$ and step $h \in [H]$.*

**Remark 5.6.** *Assumption 5.5 is a condition on the dual variable of the KL divergence, which is specific to the linear mixture DRMDP with KL divergence defined uncertainty set. We note that a similar assumption also appears in Assumption F.1 of [5], which is proposed to guarantee the provably efficient offline learning of the d-rectangular robust linear MDP.*

**Assumption 5.7** (Robust Partial Coverage: KL-divergence). *Define the worst case transition kernel $P_h^{\pi^\star,\dagger}$ and the sampling covariance matrix $\mathbf{\Lambda}_h^0$ as (5.1) and (5.3), respectively. For any $P \in \widehat{\mathcal{P}}$, denote the problem dependent robust value covariance matrix as*

$$\mathbf{\Lambda}_h^{KL}\big(\lambda; d_{h,P^{\pi^\star,\dagger}}^{\pi^\star}, V_{h+1,P}^{\pi^\star}\big) = \mathbb{E}_{(s_h,a_h)\sim d_{h,P^{\pi^\star,\dagger}}^{\pi^\star}}\Big[ \exp\big\{\frac{-\phi^{V_{h+1,P}^{\pi^\star}}}{\lambda}\big\} \exp\big\{\frac{-\phi^{V_{h+1,P}^{\pi^\star}}}{\lambda}\big\}^\top \Big]. \quad (5.4)$$

*Then we assume there exists a positive constant $C^{\pi^\star} > 0$, such that*

$$\sup_{\lambda \in [\underline{\lambda}, H/\rho]} \sup_{P \in \widehat{\mathcal{P}}} \sup_{x \in \mathbb{R}^d} \frac{x^\top \mathbf{\Lambda}_h^{KL}\big(\lambda; d_{h,P^{\pi^\star,\dagger}}^{\pi^\star}, V_{h+1,P}^{\pi^\star}\big) x}{x^\top \mathbf{\Lambda}_h^0 x} \leq \frac{C^{\pi^\star}}{K}.$$

**Remark 5.8.** *Assumption 5.7 shares the same spirit with Assumption 5.1, except that it is designed for the KL divergence based uncertainty set, which leads to the different definition of the robust value covariance matrix $\mathbf{\Lambda}_h^{KL}$, different supremum operation over the dual variable $\lambda$, and different order of the partial coverage upper bound $C^{\pi^\star}/K$.*

**Theorem 5.9** (KL-divergence). *Assume Assumptions 3.1, 4.1, 5.5 and 5.7 hold. For $\zeta \in (0,1)$, there exists an absolute constant $c > 0$, such that if we set $\lambda = d$ in Algorithm 1, then for KL-divergence uncertainty set, with probability at least $1 - \zeta$, we have*

$$SubOpt(\hat{\pi}, s_1) \leq cdH^2 C^{\pi^\star} e^{H/\underline{\lambda}} \rho^{-1} K^{-1/2} \log(K/p_{\min}d^2\zeta).$$

The above bound on suboptimality gap matches that of Theorem 1 in [2] in terms of feature dimension $d$ and horizon length $H$. However, it has an additional exponentially large term $e^{H/\underline{\lambda}}$ in the numerator and an additional $\rho$ in the denominator. Both of them are expected as they stand for the unique characteristics of DRMDP with KL divergence defined uncertainty set. Similar terms have also appeared in previous literature on tabular DRMDPs with KL divergence defined $(s, a)$-rectangular uncertainty set (Proposition 4.8 of [5]) and $d$-rectangular linear robust regularized MDPs with KL divergence regularization (Theorem 5.1 of [42]). This reflects the hardness in learning robust policies for DRMDPs with KL divergence defined uncertainty sets. Next, we translate the suboptimality bound in Theorem 5.9 to the sample complexity bound.

**Corollary 5.10.** *Under the same assumptions and setting as in Theorem 5.9, to learn an $\epsilon$-optimal policy with probability at least $1 - \zeta$, we require the sample size $K$ satisfying*

$$K = \tilde{O}\left(\frac{d^2 H^4 (C^{\pi^\star})^2 e^{2H/\underline{\lambda}}}{\rho^2 \epsilon^2}\right).$$

## 5.3 $\chi^2$-divergence

For linear mixture DRMDPs with $\chi^2$-divergence defined uncertainty sets, we additionally introduce the following coverage assumption on the offline dataset.

**Assumption 5.11** (Robust Partial Coverage: $\chi^2$-divergence). *Define the worst case transition kernel $P_h^{\pi^\star,\dagger}$ and the sampling covariance matrix $\mathbf{\Lambda}_h^0$ as (5.1) and (5.3), respectively. For any $P \in \widehat{\mathcal{P}}$, denote problem dependent robust value covariance matrix as*

$$\mathbf{\Lambda}_h^{\chi^2}\left(\alpha; d_{h,P^{\pi^\star},\dagger}^{\pi^\star}, V_{h+1,P}^{\pi^\star}\right) = \mathbb{E}_{(s_h,a_h)\sim d_{h,P^{\pi^\star},\dagger}^{\pi^\star}} \left[\left[\phi^{V_{h+1,P}^{\pi^\star}}(s_h, a_h)\right]_\alpha^2 \left[\phi^{V_{h+1,P}^{\pi^\star}}(s_h, a_h)\right]_\alpha^{2,\top}\right]. \quad (5.5)$$

*Then we assume there exists a positive constant $C^{\pi^\star} > 0$, such that*

$$\sup_{\alpha \in [0,H]} \sup_{P \in \widehat{\mathcal{P}}} \sup_{x \in \mathbb{R}^d} \frac{x^\top \mathbf{\Lambda}_h^{\chi^2}\left(\alpha; d_{h,P^{\pi^\star},\dagger}^{\pi^\star}, V_{h+1,P}^{\pi^\star}\right) x}{x^\top \mathbf{\Lambda}_h^0 x} \leq \frac{C^{\pi^\star} H^4}{K}.$$

**Theorem 5.12** ($\chi^2$-divergence). *Assume Assumptions 3.1, 4.1, 5.1 and 5.11 hold. For $\zeta \in (0,1)$, there exists an absolute constant $c > 0$, such that if we set $\lambda = d$ in Algorithm 1, then for $\chi^2$-divergence uncertainty set, with probability at least $1 - \zeta$, we have*

$$SubOpt(\hat{\pi}, s_1) \leq cd(\sqrt{\rho}H^3 + H^2)C^{\pi^\star} K^{-1/2} \log(K/p_{\min}d^2\zeta).$$

When $\rho = O(1/H^2)$, basically saying that the dynamics perturbation is negligible, the bound in Theorem 5.12 matches that for non-robust linear mixture MDP (Theorem 1 of [2]) in terms of $d$ and $H$. This makes sense as when $\rho \to 0$, the linear mixture DRMDP degrades to standard linear mixture MDP. When $\rho$ is large, our bound suggests that policy learning would be harder due to the complex geometry of the $\chi^2$ divergence uncertainty set. This aligns well with findings for $(s, a)$-rectangular tabular DRMDP with $\chi^2$ divergence defined uncertainty set (see Table 2 in [41] for details).

Next, we translate the suboptimality bound in Theorem 5.12 to the sample complexity bound.

**Corollary 5.13.** *Under the same assumptions and setting as in Theorem 5.12, to learn an $\epsilon$-optimal policy with probability at least $1 - \zeta$, we require the sample size $K$ satisfying*

$$K = \tilde{O}\left(\frac{d^2(\sqrt{\rho}H^3 + H^2)^2 (C^{\pi^\star})^2}{\epsilon^2}\right).$$

**Remark 5.14.** *The robust partial coverage assumptions Assumptions 5.1, 5.7 and 5.11 are specifically designed for linear mixture DRMDPs. They posses distinct features compared to robust partial coverage assumptions in literature, such as the robust partial coverage coefficient [40, 5] for $(s, a)$-rectangular tabular DRMDPs, and the robust partial coverage covariance matrix [5, 42] for $d$-rectangular linear robust (regularized) MDPs. Specifically, the formulation of the robust partial coverage assumption for linear Mixture DRMDPs varies according to the choice of uncertainty sets. According to the definition in (5.2), (5.4) and (5.5), different uncertainty set leads to different problem dependent robust value covariance matrix. In contrast, the robust partial coverage coefficient for tabular DRMDPs is simply defined in the form of visitation ratio, and the robust partial coverage covariance matrix for $d$-rectangular linear robust (regularized) MDPs is defined by the known feature mapping $\phi$. Both share the same form across uncertainty sets defined by different divergences.*

## 6 Discussion and conclusion

We proposed a novel framework, termed the linear mixture DRMDP, for robust policy learning. Focusing on the offline reinforcement learning setting, we introduced a meta-algorithm and formalized the assumptions necessary to study finite-sample guarantees under various instantiations of linear mixture DRMDPs. Our work lays the theoretical foundation for future research in this direction and highlights several promising avenues for exploration.

First, online learning in linear mixture DRMDPs would be an important extension, particularly relevant in applications such as robotic training with simulators. In the online setting, the agent must actively and efficiently collect data to balance the exploration–exploitation trade-off, which necessitates the design of non-trivial strategies tailored to the linear mixture DRMDP framework.

Second, learning in linear mixture DRMDPs assumes access to prior knowledge, such as the structure of the linear mixture dynamics and the basis modes. Investigating when it is appropriate to model changes in dynamics using a linear mixture uncertainty set—and how to construct meaningful basis modes in complex environments like MuJoCo—would represent an interesting future step. Several open questions arise in this context, including: How does misspecification of the basis modes affect the performance of robust policies? How do linear mixture DRMDPs compare to $(s, a)$-rectangular DRMDPs in terms of robustness and sample efficiency?

On the practical side, the proposed meta-algorithm in Algorithm 1 depends on the planning oracle defined in (4.3), rendering it computationally intractable. To demonstrate the practical utility of the linear mixture DRMDP framework, we introduced two computationally tractable algorithms in Appendix E, based on an iterative estimation subroutine, and evaluated them on simple simulated environments in Appendix F. Experimental results showed that the learned policies were robust to environment perturbations, validating the effectiveness of the proposed framework.

Nevertheless, the reliance on an iterative estimation subroutine introduces a gap between the theoretical analysis and the practical algorithms. While the primary focus of this work is to introduce the linear mixture DRMDP framework and establish its finite-sample guarantees in the offline RL setting, an important open question remains: Can we design algorithms that are both statistically and computationally efficient? We leave this as an exciting direction for future research.

## Acknowledgments

We would like to thank the anonymous reviewers for their helpful comments. ZL and PX was supported in part by the National Science Foundation (DMS-2323112) and the Whitehead Scholars Program at the Duke University School of Medicine. The views and conclusions in this paper are those of the authors and should not be interpreted as representing any funding agency.

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

# A    On practical motivation of the framework

Mixture models, in particular Gaussian Mixture Models (GMMs), are widely studied in fields of optimal control and RL. For example:

**Model Predictive Control (MPC)**    Balci and Bakolas [4] study the density steering for discrete-time linear systems, a variant of optimal mass transport problem widely used in applications such as controlling a swarm of robots/drones to achieve a desired spatial distribution. They model the stochastic dynamical system using GMMs and derive its optimal control policy. Engelaar et al. [9] study a stochastic MPC algorithm for linear systems subject to additive Gaussian mixture disturbances. They consider a vehicle control case study in which the vehicle must maintain its position on an ill-maintained road.

**Meta RL**    Meta learning is one way to increase the data efficiency of learning algorithms by generalizing learned concepts from a set of training tasks to unseen, but related, tasks. Recent works on Meta RL model the dynamics of a new system as a mixture of expert systems, thus realizing knowledge transfer. In particular, Sæmundsson et al. [37], Xu et al. [51] model the dynamics of the expert systems using Gaussian process with latent embedding. And the dynamics of the new system is constructed by a mixture of Gaussian processes with the distribution over the latent embedding being the mixture weights, which should be learned from data. In experiments, their meta-learning models effectively generalizes to novel tasks under various environments such as Cart-pole & Double-pendulum swing-up, HalfCheetah and Highway-Intersection.

**World Models for model-based RL**    Model-based Deep Reinforcement Learning (RL) assumes the availability of a model of an environment's underlying transition dynamics, which are stochastic in nature and oftentimes multimodal. Ha and Schmidhuber [15], Sedlmeier et al. [39] use Mixture-density Networks to construct World Models to capture the multimodality in dynamics. Then model-based RL methods are implemented based on the constructed World Models to solve various tasks such as car racing and Inverse Sine Wave.

The basis modes in our work can be the Gaussian components in GMMs, dynamics of expert systems, or mixture components in the Mixture-density Networks. These applications naturally give rise to distributionally robust MDP problems, for example 1) robust control of drones or vehicles is needed to hedge against environmental uncertainties and distributional perturbations; 2) given established expert systems, usually the mixture weights are estimated from limited data collected from the unseen new task, and thus of high uncertainty. Our framework could potentially enhance robustness of decision making in unseen tasks by modeling the weighting parameter uncertainty; 3) by modeling the uncertainty in the World Model (specifically, in the weights), our framework enables robust decision making in model-based RL.

# B    Proof in Section 3

In this section, we prove Lemma 3.4 and Lemma 3.5.

## B.1    Proof of Lemma 3.4

*Proof.* To see this, on the one hand, for any $(s, a) \in \mathcal{S} \times \mathcal{A}$, assume $P^0(s'|s, a) = \phi(s', s, a)^\top \theta^0$ is the nominal kernel. For any $\theta \in \{\theta \in \Delta^{d-1} | D_{\text{TV}}(\theta||\theta^0) \leq \rho\}$, we have $P(s'|s, a) = \phi(s', s, a)^\top \theta$, then

$$D_{\text{TV}}\big(P(\cdot|s, a)||P^0(\cdot|s, a)\big) = \frac{1}{2} \sum_{s' \in \mathcal{S}} \big|P(s'|s, a) - P^0(s'|s, a)\big|$$

$$= \frac{1}{2} \sum_{s' \in \mathcal{S}} \big|\phi(s', s, a)^\top (\theta - \theta^0)\big|$$

$$\leq \frac{1}{2} \sum_{s' \in \mathcal{S}} \phi(s', s, a)^\top \big|\theta - \theta^0\big|$$

$$= D_{\text{TV}}\big(\theta||\theta^0\big) \leq \rho,$$

where in the last line we used the fact that $\sum_{s' \in \mathcal{S}} \phi_i(s', s, a) = 1$ for all $i, s, a$. Thus, whenever $P$ lies in the linear mixture uncertainty set, it must also lie in the $(s,a)$-rectangular uncertainty set with the same radius. Consequently, the linear mixture uncertainty set is contained in the $(s,a)$-rectangular uncertainty set.

$\square$

## B.2 Proof of Lemma 3.5

*Proof.* To see this, we consider the tabular setting where $|\mathcal{S}|$ and $|\mathcal{A}|$ are finite, and a slight modification on Assumption 3.1. In particular, let $d = |\mathcal{S}||\mathcal{A}||\mathcal{S}|$, and $\sigma(\cdot, \cdot, \cdot) : \mathcal{S} \times \mathcal{A} \times \mathcal{S} \to [d]$ be the function that maps the state-action-next-state tuple $(s, a, s')$ to its index in the space $\mathcal{S} \times \mathcal{A} \times \mathcal{S}$. Letting

$$\phi_i(s'|s, a) = \begin{cases} 1 & \text{if } \sigma(s, a, s') = i, \\ 0 & \text{otherwise,} \end{cases}$$

and $\theta_i^0 = P^0(s'|s, a)$ if $\sigma(s, a, s') = i$. Then we have $P^0(s'|s, a) = \langle \phi(s'|s, a), \theta^0 \rangle$. To define the uncertainty set, fix any $(s, a) \in \mathcal{S} \times \mathcal{A}$, we define $\theta^0(s, a) \in \mathbb{R}^{|\mathcal{S}|}$ as the segment of $\theta^0$ corresponding to $(s, a)$. For any $\xi \in \Delta^{d-1}$, we define $\theta^0(s, a; \xi)$ as the vector of replacing the segment $\theta^0(s, a)$ in $\theta^0$ by $\xi$. Then we define the uncertainty set $\Theta(s, a) = \{\xi \in \Delta^{|\mathcal{S}|-1} | D(\xi || \theta^0(s, a)) \le \rho\}$ and $\mathcal{U}^\rho(s, a; \theta^0) = \{\theta^0(s, a; \xi) | \xi \in \Theta(s, a)\}$. Then by the construction of basis latent modes, the linear mixture uncertainty set $\mathcal{U}^\rho(P^0) = \otimes_{(s,a) \in \mathcal{S} \times \mathcal{A}} \mathcal{U}^\rho(s, a; \xi)$ is exactly the standard $(s, a)$-rectangular uncertainty set. For simplicity, we focus on the linear mixture DRMDP defined in Assumption 3.1 in the main context.

Next, we provide some cases where there exist distributions in the $(s, a)$-rectangular uncertainty set that are not in the linear mixture uncertainty set. Let's consider two simple examples.

**Example B.1.** *Let's assume $\{\phi_i(s', s, a)\}_{i=1}^d$ are identical. Then it is trivially to check that the linear mixture uncertainty set contains only the nominal kernel since there is only one basis mode and the perturbation on the weighting parameter $\theta$ does not result in a different probability distribution. While the $(s, a)$-rectangular uncertainty set defined based on the nominal distribution certainly contains more distributions other than the nominal one.*

Apart from the above degenerated example, we show another example as follows.

**Example B.2.** *Let's define $\mathcal{S} = \{x_1, x_2, x_3\}$, $d = 2$, $\theta^0 = [1/2, 1/2]^\top$, and*

$$\phi_1(\cdot) = 0.7\delta_{x_1}(\cdot) + 0.1\delta_{x_2}(\cdot) + 0.2\delta_{x_3}(\cdot),$$
$$\phi_2(\cdot) = 0.1\delta_{x_1}(\cdot) + 0.7\delta_{x_2}(\cdot) + 0.2\delta_{x_3}(\cdot),$$
$$P^0 = 0.4\delta_{x_1}(\cdot) + 0.4\delta_{x_2}(\cdot) + 0.2\delta_{x_3}(\cdot).$$

Note that $P^0 = \langle \phi, \theta \rangle$. Then for any $\rho \le \frac{1}{2}$, we have $\mathcal{U}^\rho(P^0) = (0.7\tilde{\theta}_1 + 0.1\tilde{\theta}_2)\delta_{x_1}(\cdot) + (0.1\tilde{\theta}_1 + 0.7\tilde{\theta}_2)\delta_{x_2}(\cdot) + 0.2\delta_{x_3}(\cdot)$, where $\tilde{\theta}_1 + \tilde{\theta}_2 = 1$ and $(|\theta_1^0 - \tilde{\theta}_1| + |\theta_2^0 - \tilde{\theta}_2|)/2 \le \rho$. Then for any $P \in \mathcal{U}^\rho(P^0)$ we have

$$D_{\text{TV}}(P || P^0) \le 0.8\rho,$$

and the equation can be achieved by $\theta = [1/2 + \rho, 1/2 - \rho]^\top$. Next we show that in the standard $(s, a)$-rectangularity uncertainty set around $P^0$ with radius $0.8\rho$, there exist kernels such that they are not in the linear mixture uncertainty set defined above. In particular, for any $0 < \sigma \le 0.8\rho$, we have

$$Q_\sigma = \left(0.4 - \frac{\sigma}{2}\right)\delta_{x_1} + \left(0.4 - \frac{\sigma}{2}\right)\delta_{x_2} + (0.2 - \sigma)\delta_{x_3}.$$

Since the weight of $\delta_{x_3}$ is not equal to 0.2, thus we can conclude that $Q_\sigma$ is not in the $\mathcal{U}^\rho(P^0)$. $\square$

# C  Suboptimality analysis

In this section, we prove Theorems 5.3, 5.9 and 5.12.

*Proof.* The following proof assumes that the event in Lemma 4.3 holds. Then by definition, we have

$$\begin{aligned}
\mathrm{SubOpt}(\hat{\pi}, s_1, \rho) &= V_{1,P^0}^{\pi^\star,\rho}(s_1) - V_{1,P^0}^{\hat{\pi},\rho}(s_1) \\
&= V_{1,P^0}^{\pi^\star,\rho}(s_1) - \inf_{P \in \widehat{\mathcal{P}}} V_{1,P}^{\pi^\star,\rho}(s_1) + \inf_{P \in \widehat{\mathcal{P}}} V_{1,P}^{\pi^\star,\rho}(s_1) - V_{1,P^0}^{\hat{\pi},\rho}(s_1) \\
&\leq V_{1,P^0}^{\pi^\star,\rho}(s_1) - \inf_{P \in \widehat{\mathcal{P}}} V_{1,P}^{\pi^\star,\rho}(s_1) + \inf_{P \in \widehat{\mathcal{P}}} V_{1,P}^{\hat{\pi},\rho}(s_1) - V_{1,P^0}^{\hat{\pi},\rho}(s_1) \\
&\leq V_{1,P^0}^{\pi^\star,\rho}(s_1) - \inf_{P \in \widehat{\mathcal{P}}} V_{1,P}^{\pi^\star,\rho}(s_1) \\
&= \sup_{P \in \widehat{\mathcal{P}}} \left\{ V_{1,P^0}^{\pi^\star,\rho}(s_1) - V_{1,P}^{\pi^\star,\rho}(s_1) \right\}.
\end{aligned}$$

For any $P \in \tilde{\mathcal{P}}$ and any step $h \in [H]$, we denote that

$$\begin{aligned}
\Delta_{h,P}^{\rho}(s_h, a_h) &= Q_{h,P^0}^{\pi^\star,\rho}(s_h, a_h) - Q_{h,P}^{\pi^\star,\rho}(s_h, a_h) \\
&= \inf_{\tilde{P}_h \in \mathcal{U}^\rho(P_h^0)} \mathbb{E}_{s' \sim \tilde{P}_h(\cdot|s_h,a_h)} \left[ V_{h+1,P^0}^{\pi^\star,\rho}(s') \right] - \inf_{\tilde{P}_h \in \mathcal{U}^\rho(P_h)} \mathbb{E}_{s' \sim \tilde{P}_h(\cdot|s_h,a_h)} \left[ V_{h+1,P}^{\pi^\star,\rho}(s') \right] \\
&= \underbrace{\inf_{\tilde{P}_h \in \mathcal{U}^\rho(P_h^0)} \mathbb{E}_{s' \sim \tilde{P}_h(\cdot|s_h,a_h)} \left[ V_{h+1,P^0}^{\pi^\star,\rho}(s') \right] - \inf_{\tilde{P}_h \in \mathcal{U}^\rho(P_h^0)} \mathbb{E}_{s' \sim \tilde{P}_h(\cdot|s_h,a_h)} \left[ V_{h+1,P}^{\pi^\star,\rho}(s') \right]}_{\mathrm{I}} \\
&\quad + \underbrace{\inf_{\tilde{P}_h \in \mathcal{U}^\rho(P_h^0)} \mathbb{E}_{s' \sim \tilde{P}_h(\cdot|s_h,a_h)} \left[ V_{h+1,P}^{\pi^\star,\rho}(s') \right] - \inf_{\tilde{P}_h \in \mathcal{U}^\rho(P_h)} \mathbb{E}_{s' \sim \tilde{P}_h(\cdot|s_h,a_h)} \left[ V_{h+1,P}^{\pi^\star,\rho}(s') \right]}_{\mathrm{II}}.
\end{aligned}$$

For term I, define

$$P_h^{\pi^\star,\dagger} = \underset{\tilde{P}_h \in \mathcal{U}^\rho(P_h^0)}{\arg\inf} \mathbb{E}_{s' \sim \tilde{P}_h(\cdot|s,a)} \left[ V_{h+1,P}^{\pi^\star,\rho}(s') \right] \quad \forall (s, a) \in \mathcal{S} \times \mathcal{A}.$$

Then we have

$$\begin{aligned}
\mathrm{I} &\leq \mathbb{E}_{s' \sim P_h^{\pi^\star,\dagger}(\cdot|s_h,a_h)} \left[ V_{h+1,P^0}^{\pi^\star,\rho}(s') \right] - \mathbb{E}_{s' \sim P_h^{\pi^\star,\dagger}(\cdot|s_h,a_h)} \left[ V_{h+1,P}^{\pi^\star,\rho}(s') \right] \\
&= \mathbb{E}_{s' \sim P_h^{\pi^\star,\dagger}(\cdot|s_h,a_h),a' \sim \pi_{h+1}^\star(\cdot|s')} \left[ \Delta_{h+1,P}^{\rho}(s', a') \right].
\end{aligned}$$

For the term II, we denote it by $\Delta_{h,P}^{(\mathrm{II})}(s_h, a_h)$ for simplicity. Then we have

$$\Delta_{h,P}^{\rho}(s_h, a_h) = \mathrm{I} + \mathrm{II} \leq \mathbb{E}_{s' \sim P_h^{\pi^\star,\dagger}(\cdot|s_h,a_h),a' \sim \pi_{h+1}^\star(\cdot|s')} \left[ \Delta_{h+1,P}^{\rho}(s', a') \right] + \Delta_{h,P}^{(\mathrm{II})}(s_h, a_h). \quad \text{(C.1)}$$

Recursively applying (C.1) and the plugging in the definition of $\Delta_{h,P}^{(\mathrm{II})}(s_h, a_h)$, we can obtain that

$$\begin{aligned}
&\mathbb{E}_{a_1 \sim \pi_1^\star(\cdot|s_1)} \left[ \Delta_{1,P}^{\rho}(s_1, a_1) \right] \\
&\leq \sum_{h=1}^{H} \mathbb{E}_{(s_h,a_h) \sim d_{h,P_h^{\pi^\star,\dagger}}^{\pi^\star}} \left[ \inf_{\tilde{P}_h \in \mathcal{U}^\rho(P_h^0)} \mathbb{E}_{s' \sim \tilde{P}_h(\cdot|s_h,a_h)} \left[ V_{h+1,P}^{\pi^\star,\rho}(s') \right] - \inf_{\tilde{P}_h \in \mathcal{U}^\rho(P_h)} \mathbb{E}_{s' \sim \tilde{P}_h(\cdot|s_h,a_h)} \left[ V_{h+1,P}^{\pi^\star,\rho}(s') \right] \right]
\end{aligned}$$

$$\text{(C.2)}$$

Next, we study

$$\inf_{\tilde{P}_h \in \mathcal{U}^\rho(P_h^0)} \mathbb{E}_{s' \sim \tilde{P}_h(\cdot|s_h,a_h)} \left[ V_{h+1,P}^{\pi^\star,\rho}(s') \right] - \inf_{\tilde{P}_h \in \mathcal{U}^\rho(P_h)} \mathbb{E}_{s' \sim \tilde{P}_h(\cdot|s_h,a_h)} \left[ V_{h+1,P}^{\pi^\star,\rho}(s') \right]$$

under different kinds of divergences.

**I. TV-divergence**  Let $\mathcal{U}^\rho$ defined by the TV-divergence, we have

$$
\inf_{\tilde{P}_h \in \mathcal{U}^\rho(P_h^0)} \mathbb{E}_{s' \sim \tilde{P}_h(\cdot|s_h,a_h)} \left[ V_{h+1,P}^{\pi^\star,\rho}(s') \right] - \inf_{\tilde{P}_h \in \mathcal{U}^\rho(P_h)} \mathbb{E}_{s' \sim \tilde{P}_h(\cdot|s_h,a_h)} \left[ V_{h+1,P}^{\pi^\star,\rho}(s') \right]
$$

$$
= \inf_{\tilde{\boldsymbol{\theta}}_h \in \mathcal{U}^\rho(\boldsymbol{\theta}_h^0)} \mathbb{E}_{i \sim \tilde{\boldsymbol{\theta}}_h} \left[ \phi_i^{V_{h+1,P}^{\pi^\star,\rho}}(s_h,a_h) \right] - \inf_{\tilde{\boldsymbol{\theta}}_h \in \mathcal{U}^\rho(\boldsymbol{\theta}_h)} \mathbb{E}_{i \sim \tilde{\boldsymbol{\theta}}_h} \left[ \phi_i^{V_{h+1,P}^{\pi^\star,\rho}}(s_h,a_h) \right]
$$

$$
= \sup_{\alpha \in [0,H]} \left\{ \mathbb{E}_{i \sim \boldsymbol{\theta}_h^0} \left[ \phi_i^{V_{h+1,P}^{\pi^\star,\rho}}(s_h,a_h) \right]_\alpha - \rho \left( \alpha - \min_i \left[ \phi_i^{V_{h+1,P}^{\pi^\star,\rho}}(s_h,a_h) \right]_\alpha \right) \right\}
$$

$$
- \sup_{\alpha \in [0,H]} \left\{ \mathbb{E}_{i \sim \boldsymbol{\theta}_h} \left[ \phi_i^{V_{h+1,P}^{\pi^\star,\rho}}(s_h,a_h) \right]_\alpha - \rho \left( \alpha - \min_i \left[ \phi_i^{V_{h+1,P}^{\pi^\star,\rho}}(s_h,a_h) \right]_\alpha \right) \right\}
$$

$$
\leq \sup_{\alpha \in [0,H]} \left\{ \left( \mathbb{E}_{i \sim \boldsymbol{\theta}_h^0} - \mathbb{E}_{i \sim \boldsymbol{\theta}_h} \right) \left[ \phi_i^{V_{h+1,P}^{\pi^\star,\rho}}(s_h,a_h) \right]_\alpha \right\}.
$$

Denote

$$
\alpha_h^\star = \arg\sup_{\alpha \in [0,H]} \left\{ \left( \mathbb{E}_{i \sim \boldsymbol{\theta}_h^0} - \mathbb{E}_{i \sim \boldsymbol{\theta}_h} \right) \left[ \phi_i^{V_{h+1,P}^{\pi^\star,\rho}}(s_h,a_h) \right]_\alpha \right\},
$$

then we have

$$
\inf_{\tilde{P}_h \in \mathcal{U}^\rho(P_h^0)} \mathbb{E}_{s' \sim \tilde{P}_h(\cdot|s_h,a_h)} \left[ V_{h+1,P}^{\pi^\star,\rho}(s') \right] - \inf_{\tilde{P}_h \in \mathcal{U}^\rho(P_h)} \mathbb{E}_{s' \sim \tilde{P}_h(\cdot|s_h,a_h)} \left[ V_{h+1,P}^{\pi^\star,\rho}(s') \right]
$$

$$
\leq \left( \mathbb{E}_{i \sim \boldsymbol{\theta}_h^0} - \mathbb{E}_{i \sim \boldsymbol{\theta}_h} \right) \left[ \phi_i^{V_{h+1,P}^{\pi^\star,\rho}}(s_h,a_h) \right]_{\alpha_h^\star}
$$

$$
= \left\langle \boldsymbol{\theta}_h^0 - \boldsymbol{\theta}_h, \left[ \boldsymbol{\phi}^{V_{h+1,P}^{\pi^\star,\rho}}(s_h,a_h) \right]_{\alpha_h^\star} \right\rangle
$$

$$
\leq \| \boldsymbol{\theta}_h^0 - \boldsymbol{\theta}_h \|_{\boldsymbol{\Lambda}_h} \left\| \left[ \boldsymbol{\phi}^{V_{h+1,P}^{\pi^\star,\rho}}(s_h,a_h) \right]_{\alpha_h^\star} \right\|_{\boldsymbol{\Lambda}_h^{-1}} \tag{C.3}
$$

Combining (C.2) and (C.3), we have

$$
\mathbb{E}_{a_1 \sim \pi_1^\star(\cdot|s_1)} \left[ \Delta_{1,P}^\rho(s_1,a_1) \right]
$$

$$
\leq \sum_{h=1}^{H} \mathbb{E}_{(s_h,a_h) \sim d_{h,P^{\pi^\star}}^{\pi^\star,\dagger}} \left[ \| \boldsymbol{\theta}_h^0 - \boldsymbol{\theta}_h \|_{\boldsymbol{\Lambda}_h} \left\| \left[ \boldsymbol{\phi}^{V_{h+1,P}^{\pi^\star,\rho}}(s_h,a_h) \right]_{\alpha_h^\star} \right\|_{\boldsymbol{\Lambda}_h^{-1}} \right]
$$

$$
\leq \sum_{h=1}^{H} \beta_h \left( \mathbb{E}_{(s_h,a_h) \sim d_{h,P^{\pi^\star}}^{\pi^\star,\dagger}} \left[ \left\| \left[ \boldsymbol{\phi}^{V_{h+1,P}^{\pi^\star,\rho}}(s_h,a_h) \right]_{\alpha_h^\star} \right\|_{\boldsymbol{\Lambda}_h^{-1}} \right] \right)^{1/2} \tag{C.4}
$$

$$
\leq \beta_h \sqrt{ \text{Tr} \left( \mathbb{E}_{(s_h,a_h) \sim d_{h,P^{\pi^\star}}^{\pi^\star,\dagger}} \left[ \left[ \boldsymbol{\phi}^{V_{h+1,P}^{\pi^\star,\rho}}(s_h,a_h) \right]_{\alpha_h^\star} \left[ \boldsymbol{\phi}^{V_{h+1,P}^{\pi^\star,\rho}}(s_h,a_h) \right]_{\alpha_h^\star}^\top \right] \boldsymbol{\Lambda}_h^{-1} \right) }
$$

$$
= \sum_{h=1}^{H} \beta_h \sqrt{ \text{Tr} \left( \boldsymbol{\Lambda}_h^{\text{TV}}(\alpha_h^\star; d_{h,P^{\pi^\star},\dagger}^{\pi^\star}, V_{h+1,P}^{\pi^\star,\rho}) \boldsymbol{\Lambda}_h^{-1} \right) }
$$

$$
\leq \sum_{h=1}^{H} \beta_h \sqrt{ \sup_{x \in \mathbb{R}^d} \frac{x^\top \boldsymbol{\Lambda}_h^{\text{TV}}(\alpha_h^\star; d_{h,P^{\pi^\star},\dagger}^{\pi^\star}, V_{h+1,P}^{\pi^\star,\rho}) x}{x^\top \boldsymbol{\Lambda}_h^0 x} \text{Tr} \left( \boldsymbol{\Lambda}_h^0 \boldsymbol{\Lambda}_h^{-1} \right) } \tag{C.5}
$$

$$
\leq \sum_{h=1}^{H} \beta_h \sqrt{ \frac{1}{K} \cdot H^2 \cdot C^{\pi^\star} \cdot \text{Rank}(\boldsymbol{\Lambda}_h^0) } \tag{C.6}
$$

$$
= \sum_{h=1}^{H} \beta_h \sqrt{ \frac{1}{K} \cdot H^2 \cdot C^{\pi^\star} \cdot d }
$$

$$
= \frac{c H^2 d C^{\pi^\star} \log(K/d^2 p_{\min} \zeta)}{\sqrt{K}}, \tag{C.7}
$$

where (C.4) holds due to Jensen's inequality, (C.5) holds by Lemma D.3 and the fact that $\mathbf{\Lambda}_h^0 \preceq \mathbf{\Lambda}_h$, (C.6) holds by Assumption 5.1, and (C.7) holds by the fact $\lambda = d$ and bounding $\beta_h$ as follows

$$\beta_h = \frac{5}{4}\sqrt{\lambda} + \frac{2}{\sqrt{\lambda}}\left(2\log\frac{H}{\zeta} + d\log\left(4 + \frac{4\lceil 1/p_{\min}\rceil K}{\lambda d}\right)\right) \leq c\sqrt{d}\log\frac{K}{p_{\min}d^2\zeta}.$$

Thus, we have

$$\mathrm{SubOpt}(\hat{\pi}, s_1, \rho) \leq \frac{cdH^2C^{\pi^\star}\log(K/d^2p_{\min}\zeta)}{\sqrt{K}}.$$

We complete the proof of Theorem 5.3.

## II. KL-divergence

$$\inf_{\tilde{P}_h \in \mathcal{U}^\rho(P_h^0)} \mathbb{E}_{s'\sim\tilde{P}_h(\cdot|s_h,a_h)}\left[V_{h+1,P}^{\pi^\star,\rho}(s')\right] - \inf_{\tilde{P}_h \in \mathcal{U}^\rho(P_h)} \mathbb{E}_{s'\sim\tilde{P}_h(\cdot|s_h,a_h)}\left[V_{h+1,P}^{\pi^\star,\rho}(s')\right]$$

$$= \inf_{\tilde{\boldsymbol{\theta}}_h \in \mathcal{U}^\rho(\boldsymbol{\theta}_h^0)} \mathbb{E}_{i\sim\tilde{\boldsymbol{\theta}}_h}\left[\phi_i^{V_{h+1,P}^{\pi^\star,\rho}}(s_h,a_h)\right] - \inf_{\tilde{\boldsymbol{\theta}}_h \in \mathcal{U}^\rho(\boldsymbol{\theta}_h)} \mathbb{E}_{i\sim\tilde{\boldsymbol{\theta}}_h}\left[\phi_i^{V_{h+1,P}^{\pi^\star,\rho}}(s_h,a_h)\right]$$

$$= \sup_{\lambda\in[\underline{\lambda},H/\rho]}\left\{-\lambda\log\left(\mathbb{E}_{i\sim\boldsymbol{\theta}_h^0}\left[\exp\left\{-\phi_i^{V_{h+1,P}^{\pi^\star,\rho}}(s_h,a_h)/\lambda\right\}\right]\right) - \lambda\rho\right\}$$

$$\quad - \sup_{\lambda\in[\underline{\lambda},H/\rho]}\left\{-\lambda\log\left(\mathbb{E}_{i\sim\boldsymbol{\theta}_h}\left[\exp\left\{-\phi_i^{V_{h+1,P}^{\pi^\star,\rho}}(s_h,a_h)/\lambda\right\}\right]\right) - \lambda\rho\right\}$$

$$\leq \sup_{\lambda\in[\underline{\lambda},H/\rho]}\left\{\lambda\log\left(\frac{\mathbb{E}_{i\sim\boldsymbol{\theta}_h}\exp\left\{-\phi_i^{V_{h+1,P}^{\pi^\star,\rho}}(s_h,a_h)/\lambda\right\}}{\mathbb{E}_{i\sim\boldsymbol{\theta}_h^0}\exp\left\{-\phi_i^{V_{h+1,P}^{\pi^\star,\rho}}(s_h,a_h)/\lambda\right\}}\right)\right\}.$$

Denote

$$\lambda_h^\star = \arg\sup_{\lambda\in[\underline{\lambda},H/\rho]}\left\{\lambda\log\left(\frac{\mathbb{E}_{i\sim\boldsymbol{\theta}_h}\exp\left\{-\phi_i^{V_{h+1,P}^{\pi^\star,\rho}}(s_h,a_h)/\lambda\right\}}{\mathbb{E}_{i\sim\boldsymbol{\theta}_h^0}\exp\left\{-\phi_i^{V_{h+1,P}^{\pi^\star,\rho}}(s_h,a_h)/\lambda\right\}}\right)\right\}.$$

Then we have

$$\inf_{\tilde{P}_h \in \mathcal{U}^\rho(P_h^0)} \mathbb{E}_{s'\sim\tilde{P}_h(\cdot|s_h,a_h)}\left[V_{h+1,P}^{\pi^\star,\rho}(s')\right] - \inf_{\tilde{P}_h \in \mathcal{U}^\rho(P_h)} \mathbb{E}_{s'\sim\tilde{P}_h(\cdot|s_h,a_h)}\left[V_{h+1,P}^{\pi^\star,\rho}(s')\right]$$

$$\leq \lambda_h^\star\log\left(\frac{\mathbb{E}_{i\sim\boldsymbol{\theta}_h}\exp\left\{-\phi_i^{V_{h+1,P}^{\pi^\star,\rho}}(s_h,a_h)/\lambda_h^\star\right\}}{\mathbb{E}_{i\sim\boldsymbol{\theta}_h^0}\exp\left\{-\phi_i^{V_{h+1,P}^{\pi^\star,\rho}}(s_h,a_h)/\lambda_h^\star\right\}}\right)$$

$$= \lambda_h^\star\log\left(1 + \frac{\left(\mathbb{E}_{i\sim\boldsymbol{\theta}_h} - \mathbb{E}_{i\sim\boldsymbol{\theta}_h^0}\right)\exp\left\{-\phi_i^{V_{h+1,P}^{\pi^\star,\rho}}(s_h,a_h)/\lambda_h^\star\right\}}{\mathbb{E}_{i\sim\boldsymbol{\theta}_h^0}\exp\left\{-\phi_i^{V_{h+1,P}^{\pi^\star,\rho}}(s_h,a_h)/\lambda_h^\star\right\}}\right)$$

$$\leq \lambda_h^\star\frac{\left(\mathbb{E}_{i\sim\boldsymbol{\theta}_h} - \mathbb{E}_{i\sim\boldsymbol{\theta}_h^0}\right)\exp\left\{-\phi_i^{V_{h+1,P}^{\pi^\star,\rho}}(s_h,a_h)/\lambda_h^\star\right\}}{\mathbb{E}_{i\sim\boldsymbol{\theta}_h^0}\exp\left\{-\phi_i^{V_{h+1,P}^{\pi^\star,\rho}}(s_h,a_h)/\lambda_h^\star\right\}}$$

$$\leq \frac{He^{H/\underline{\lambda}}}{\rho}\left|\left(\mathbb{E}_{i\sim\boldsymbol{\theta}_h} - \mathbb{E}_{i\sim\boldsymbol{\theta}_h^0}\right)\exp\left\{-\phi_i^{V_{h+1,P}^{\pi^\star,\rho}}(s_h,a_h)/\lambda_h^\star\right\}\right|$$

$$\leq \frac{He^{H/\underline{\lambda}}}{\rho}\left\|\boldsymbol{\theta}_h - \boldsymbol{\theta}_h^0\right\|_{\mathbf{\Lambda}_h} \cdot \left\|\exp\left\{-\phi^{V_{h+1,P}^{\pi^\star,\rho}}(s_h,a_h)/\lambda_h^\star\right\}\right\|_{\mathbf{\Lambda}_h^{-1}}. \tag{C.8}$$

Combining (C.2) and (C.8), we have

$$\mathbb{E}_{a_1\sim\pi_1^\star(\cdot|s_1)}\left[\Delta_{1,P}^\rho(s_1,a_1)\right]\cdot\left(\frac{He^{H/\underline{\lambda}}}{\rho}\right)^{-1}$$

$$\leq \sum_{h=1}^{H} \mathbb{E}_{(s_h,a_h)\sim d_{h,P\pi^\star,\dagger}^{\pi^\star}} \left[ \left\| \boldsymbol{\theta}_h - \boldsymbol{\theta}_h^0 \right\|_{\boldsymbol{\Lambda}_h} \cdot \left\| \exp\left\{ -\boldsymbol{\phi}^{V_{h+1,P}^{\pi^\star,\rho}}(s_h,a_h)/\lambda_h^\star \right\} \right\|_{\boldsymbol{\Lambda}_h^{-1}} \right]$$

$$\leq \sum_{h=1}^{H} \left\| \boldsymbol{\theta}_h - \boldsymbol{\theta}_h^0 \right\|_{\boldsymbol{\Lambda}_h} \cdot \left( \mathbb{E}_{(s_h,a_h)\sim d_{h,P\pi^\star,\dagger}^{\pi^\star}} \left[ \left\| \exp\left\{ -\boldsymbol{\phi}^{V_{h+1,P}^{\pi^\star,\rho}}(s_h,a_h)/\lambda_h^\star \right\} \right\|_{\boldsymbol{\Lambda}_h^{-1}}^2 \right] \right)^{1/2} \qquad \text{(C.9)}$$

$$\leq \sum_{h=1}^{H} \beta_h \sqrt{\text{Tr}\left( \mathbb{E}_{(s_h,a_h)\sim d_{h,P\pi^\star,\dagger}^{\pi^\star}} \left[ \exp\left\{ -\boldsymbol{\phi}^{V_{h+1,P}^{\pi^\star,\rho}}(s_h,a_h)/\lambda_h^\star \right\} \exp\left\{ -\boldsymbol{\phi}^{V_{h+1,P}^{\pi^\star,\rho}}(s_h,a_h)/\lambda_h^\star \right\}^\top \right] \boldsymbol{\Lambda}_h^{-1} \right)}$$

$$= \sum_{h=1}^{H} \beta_h \sqrt{\text{Tr}\left( \boldsymbol{\Lambda}_h^{\text{KL}}(\lambda_h^\star; d_{h,P\pi^\star,\dagger}^{\pi^\star}, V_{h+1,P}^{\pi^\star}) \boldsymbol{\Lambda}_h^{-1} \right)}$$

$$\leq \sum_{h=1}^{H} \beta_h \sqrt{ \sup_{x\in\mathbb{R}^d} \frac{x^\top \boldsymbol{\Lambda}_h^{\text{KL}}(\lambda_h^\star; d_{h,P\pi^\star,\dagger}^{\pi^\star}, V_{h+1,P}^{\pi^\star})x}{x^\top \boldsymbol{\Lambda}_h^0 x} \, \text{Tr}\left( \boldsymbol{\Lambda}_h^0 \boldsymbol{\Lambda}_h^{-1} \right)} \qquad \text{(C.10)}$$

$$\leq \sum_{h=1}^{H} \beta_h \sqrt{ \frac{1}{K} \cdot C^{\pi^\star} \cdot \text{Rank}(\boldsymbol{\Lambda}_h^0)} \qquad \text{(C.11)}$$

$$= \sum_{h=1}^{H} \beta_h \sqrt{ \frac{1}{K} \cdot C^{\pi^\star} \cdot d}$$

$$= \frac{cHdC^{\pi^\star}\log(K/d^2 p_{\min}\zeta)}{\sqrt{K}}, \qquad \text{(C.12)}$$

where (C.9) holds by the Jensen's inequality, (C.10) holds by Lemma D.3 and the fact that $\boldsymbol{\Lambda}_h^0 \preceq \boldsymbol{\Lambda}_h$, (C.11) holds by Assumption 5.7, and (C.12) holds by the fact $\lambda = d$ and bounding $\beta_h$ as follows

$$\beta_h = \frac{5}{4}\sqrt{\lambda} + \frac{2}{\sqrt{\lambda}}\left( 2\log\frac{H}{\zeta} + d\log\left( 4 + \frac{4\lceil 1/p_{\min}\rceil K}{\lambda d} \right) \right) \leq c\sqrt{d}\log\frac{K}{p_{\min}d^2\zeta}.$$

Thus, we have

$$\text{SubOpt}(\hat{\pi}, s_1, \rho) \leq \frac{cdH^2 C^{\pi^\star} e^{H/\underline{\lambda}}\log(K/d^2 p_{\min}\zeta)}{\sqrt{K} \cdot \rho}.$$

We complete the proof of Theorem 5.9.

### III. $\chi^2$-divergence

$$\inf_{\tilde{P}_h\in\mathcal{U}^\rho(P_h^0)} \mathbb{E}_{s'\sim\tilde{P}_h(\cdot|s_h,a_h)}\left[ V_{h+1,P}^{\pi^\star,\rho}(s') \right] - \inf_{\tilde{P}_h\in\mathcal{U}^\rho(P_h)} \mathbb{E}_{s'\sim\tilde{P}_h(\cdot|s_h,a_h)}\left[ V_{h+1,P}^{\pi^\star,\rho}(s') \right]$$

$$= \inf_{\tilde{\boldsymbol{\theta}}_h\in\mathcal{U}^\rho(\boldsymbol{\theta}_h^0)} \mathbb{E}_{i\sim\tilde{\boldsymbol{\theta}}_h}\left[ \phi_i^{V_{h+1,P}^{\pi^\star,\rho}}(s_h,a_h) \right] - \inf_{\tilde{\boldsymbol{\theta}}_h\in\mathcal{U}^\rho(\boldsymbol{\theta}_h)} \mathbb{E}_{i\sim\tilde{\boldsymbol{\theta}}_h}\left[ \phi_i^{V_{h+1,P}^{\pi^\star,\rho}}(s_h,a_h) \right]$$

$$= \sup_{\alpha\in[0,H]} \left\{ \mathbb{E}_{i\sim\boldsymbol{\theta}_h^0}\left[ \phi_i^{V_{h+1,P}^{\pi^\star,\rho}}(s_h,a_h) \right]_\alpha - \sqrt{\rho\,\text{Var}_{i\sim\boldsymbol{\theta}_h^0}\left( \left[ \phi_i^{V_{h+1,P}^{\pi^\star,\rho}}(s_h,a_h) \right]_\alpha \right)} \right\}$$

$$- \sup_{\alpha\in[0,H]} \left\{ \mathbb{E}_{i\sim\boldsymbol{\theta}_h}\left[ \phi_i^{V_{h+1,P}^{\pi^\star,\rho}}(s_h,a_h) \right]_\alpha - \sqrt{\rho\,\text{Var}_{i\sim\boldsymbol{\theta}_h}\left( \left[ \phi_i^{V_{h+1,P}^{\pi^\star,\rho}}(s_h,a_h) \right]_\alpha \right)} \right\}$$

$$\leq \sup_{\alpha\in[0,H]} \left\{ \left( \mathbb{E}_{i\sim\boldsymbol{\theta}_h^0} - \mathbb{E}_{i\sim\boldsymbol{\theta}_h} \right)\left[ \phi_i^{V_{h+1,P}^{\pi^\star,\rho}}(s_h,a_h) \right]_\alpha - \sqrt{\rho\,\text{Var}_{i\sim\boldsymbol{\theta}_h^0}\left( \left[ \phi_i^{V_{h+1,P}^{\pi^\star,\rho}}(s_h,a_h) \right]_\alpha \right)} \right.$$

$$\left. + \sqrt{\rho\,\text{Var}_{i\sim\boldsymbol{\theta}_h}\left( \left[ \phi_i^{V_{h+1,P}^{\pi^\star,\rho}}(s_h,a_h) \right]_\alpha \right)} \right\}.$$

Denote

$$\alpha_h^\star = \arg\sup_{\alpha\in[0,H]} \left\{ \left( \mathbb{E}_{i\sim\boldsymbol{\theta}_h^0} - \mathbb{E}_{i\sim\boldsymbol{\theta}_h} \right)\left[ \phi_i^{V_{h+1,P}^{\pi^\star,\rho}}(s_h,a_h) \right]_\alpha - \sqrt{\rho\,\text{Var}_{i\sim\boldsymbol{\theta}_h^0}\left( \left[ \phi_i^{V_{h+1,P}^{\pi^\star,\rho}}(s_h,a_h) \right]_\alpha \right)} \right.$$

$$+ \sqrt{\rho \operatorname{Var}_{i \sim \boldsymbol{\theta}_h} \left( \left[ \phi_i^{V_{h+1,P}^{\pi^\star,\rho}}(s_h, a_h) \right]_\alpha \right)} \Big\},$$

then we have

$$\inf_{\tilde{P}_h \in \mathcal{U}^\rho(P_h^0)} \mathbb{E}_{s' \sim \tilde{P}_h(\cdot|s_h,a_h)} \left[ V_{h+1,P}^{\pi^\star,\rho}(s') \right] - \inf_{\tilde{P}_h \in \mathcal{U}^\rho(P_h)} \mathbb{E}_{s' \sim \tilde{P}_h(\cdot|s_h,a_h)} \left[ V_{h+1,P}^{\pi^\star,\rho}(s') \right]$$

$$= \left( \mathbb{E}_{i \sim \boldsymbol{\theta}_h^0} - \mathbb{E}_{i \sim \boldsymbol{\theta}_h} \right) \left[ \phi_i^{V_{h+1,P}^{\pi^\star,\rho}}(s_h, a_h) \right]_{\alpha_h^\star} - \sqrt{\rho \operatorname{Var}_{i \sim \boldsymbol{\theta}_h^0} \left( \left[ \phi_i^{V_{h+1,P}^{\pi^\star,\rho}}(s_h, a_h) \right]_{\alpha_h^\star} \right)}$$

$$+ \sqrt{\rho \operatorname{Var}_{i \sim \boldsymbol{\theta}_h} \left( \left[ \phi_i^{V_{h+1,P}^{\pi^\star,\rho}}(s_h, a_h) \right]_{\alpha_h^\star} \right)}$$

$$\leq \left\| \boldsymbol{\theta}_h - \boldsymbol{\theta}_h^0 \right\|_{\boldsymbol{\Lambda}_h} \cdot \left\| \left[ \phi_i^{V_{h+1,P}^{\pi^\star,\rho}}(s_h, a_h) \right]_{\alpha_h^\star} \right\|_{\boldsymbol{\Lambda}_h^{-1}}$$

$$+ \sqrt{\left| \rho \operatorname{Var}_{i \sim \boldsymbol{\theta}_h} \left( \left[ \phi_i^{V_{h+1,P}^{\pi^\star,\rho}}(s_h, a_h) \right]_{\alpha_h^\star} \right) - \rho \operatorname{Var}_{i \sim \boldsymbol{\theta}_h^0} \left( \left[ \phi_i^{V_{h+1,P}^{\pi^\star,\rho}}(s_h, a_h) \right]_{\alpha_h^\star} \right) \right|}$$

$$= \left\| \boldsymbol{\theta}_h - \boldsymbol{\theta}_h^0 \right\|_{\boldsymbol{\Lambda}_h} \cdot \left\| \left[ \phi_i^{V_{h+1,P}^{\pi^\star,\rho}}(s_h, a_h) \right]_{\alpha_h^\star} \right\|_{\boldsymbol{\Lambda}_h^{-1}}$$

$$+ \sqrt{\rho} \left( \mathbb{E}_{i \sim \boldsymbol{\theta}_h^0} \left[ \phi_i^{V_{h+1,P}^{\pi^\star,\rho}}(s_h, a_h) \right]_{\alpha_h^\star}^2 - \left( \mathbb{E}_{i \sim \boldsymbol{\theta}_h^0} \left[ \phi_i^{V_{h+1,P}^{\pi^\star,\rho}}(s_h, a_h) \right]_{\alpha_h^\star} \right)^2 \right.$$

$$\left. - \mathbb{E}_{i \sim \boldsymbol{\theta}_h} \left[ \phi_i^{V_{h+1,P}^{\pi^\star,\rho}}(s_h, a_h) \right]_{\alpha_h^\star}^2 + \left( \mathbb{E}_{i \sim \boldsymbol{\theta}_h} \left[ \phi_i^{V_{h+1,P}^{\pi^\star,\rho}}(s_h, a_h) \right]_{\alpha_h^\star} \right)^2 \right)^{1/2}$$

$$\leq \left\| \boldsymbol{\theta}_h - \boldsymbol{\theta}_h^0 \right\|_{\boldsymbol{\Lambda}_h} \cdot \left\| \left[ \phi_i^{V_{h+1,P}^{\pi^\star,\rho}}(s_h, a_h) \right]_{\alpha_h^\star} \right\|_{\boldsymbol{\Lambda}_h^{-1}}$$

$$+ \sqrt{\rho} \sqrt{\mathbb{E}_{i \sim \boldsymbol{\theta}_h^0} \left[ \phi_i^{V_{h+1,P}^{\pi^\star,\rho}}(s_h, a_h) \right]_{\alpha_h^\star}^2 - \mathbb{E}_{i \sim \boldsymbol{\theta}_h} \left[ \phi_i^{V_{h+1,P}^{\pi^\star,\rho}}(s_h, a_h) \right]_{\alpha_h^\star}^2}$$

$$+ \sqrt{\rho} \sqrt{\left( \mathbb{E}_{i \sim \boldsymbol{\theta}_h^0} \left[ \phi_i^{V_{h+1,P}^{\pi^\star,\rho}}(s_h, a_h) \right]_{\alpha_h^\star} \right)^2 - \left( \mathbb{E}_{i \sim \boldsymbol{\theta}_h} \left[ \phi_i^{V_{h+1,P}^{\pi^\star,\rho}}(s_h, a_h) \right]_{\alpha_h^\star} \right)^2}$$

$$\leq \underbrace{\left\| \boldsymbol{\theta}_h - \boldsymbol{\theta}_h^0 \right\|_{\boldsymbol{\Lambda}_h} \cdot \left\| \left[ \phi_i^{V_{h+1,P}^{\pi^\star,\rho}}(s_h, a_h) \right]_{\alpha_h^\star} \right\|_{\boldsymbol{\Lambda}_h^{-1}}}_{\mathrm{I}_h} + \underbrace{\sqrt{\rho} \sqrt{\left\| \boldsymbol{\theta}_h - \boldsymbol{\theta}_h^0 \right\|_{\boldsymbol{\Lambda}_h} \cdot \left\| \left[ \phi_i^{V_{h+1,P}^{\pi^\star,\rho}}(s_h, a_h) \right]_{\alpha_h^\star}^2 \right\|_{\boldsymbol{\Lambda}_h^{-1}}}}_{\mathrm{II}_h}$$

$$\tag{C.13}$$

$$+ \underbrace{\sqrt{\rho} \sqrt{2H \left\| \boldsymbol{\theta}_h - \boldsymbol{\theta}_h^0 \right\|_{\boldsymbol{\Lambda}_h} \cdot \left\| \left[ \phi_i^{V_{h+1,P}^{\pi^\star,\rho}}(s_h, a_h) \right]_{\alpha_h^\star} \right\|_{\boldsymbol{\Lambda}_h^{-1}}}}_{\mathrm{III}_h} \tag{C.14}$$

Combining (C.2) and (C.14), then we have

$$\mathbb{E}_{a_1 \sim \pi_1^\star(\cdot|s_1)} \left[ \Delta_{1,P}^\rho(s_1, a_1) \right] \leq \sum_{h=1}^H \mathbb{E}_{(s_h,a_h) \sim d_{h,P^{\pi^\star,\dagger}}^{\pi^\star}} \left[ \mathrm{I}_h + \mathrm{II}_h + \mathrm{III}_h \right].$$

Note that, by the similar proof as that of the Case I. TV-divergence, we immediately have

$$\sum_{h=1}^H \mathbb{E}_{(s_h,a_h) \sim d_{h,P^{\pi^\star,\dagger}}^{\pi^\star}} \left[ \mathrm{I}_h + \mathrm{III}_h \right] \leq \frac{cH^2 dC^{\pi^\star} \log(K/d^2 p_{\min}\zeta)}{\sqrt{K}} + \frac{c\sqrt{\rho}H^{5/2} dC^{\pi^\star} \log(K/d^2 p_{\min}\zeta)}{\sqrt{K}}.$$

Next, we study

$$\sum_{h=1}^H \mathbb{E}_{(s_h,a_h) \sim d_{h,P^{\pi^\star,\dagger}}^{\pi^\star}} \left[ \mathrm{II}_h \right]$$

$$= \sqrt{\rho} \sum_{h=1}^{H} \mathbb{E}_{(s_h,a_h) \sim d_{h,P\pi^\star,\dagger}^{\pi^\star}} \sqrt{\left\| \boldsymbol{\theta}_h - \boldsymbol{\theta}_h^0 \right\|_{\boldsymbol{\Lambda}_h} \cdot \left\| \left[ \phi_i^{V_{h+1,P}^{\pi^\star,\rho}}(s_h,a_h) \right]_{\alpha_h^\star}^2 \right\|_{\boldsymbol{\Lambda}_h^{-1}}}$$

$$\leq \sqrt{\rho} \sum_{h=1}^{H} \beta_h \left( \mathbb{E}_{(s_h,a_h) \sim d_{h,P\pi^\star,\dagger}^{\pi^\star}} \left[ \left\| \left[ \phi_i^{V_{h+1,P}^{\pi^\star,\rho}}(s_h,a_h) \right]_{\alpha_h^\star}^2 \right\|_{\boldsymbol{\Lambda}_h^{-1}}^2 \right] \right)^{1/2} \tag{C.15}$$

$$\leq \sqrt{\rho} \sum_{h=1}^{H} \beta_h \sqrt{\mathrm{Tr}\left( \mathbb{E}_{(s_h,a_h) \sim d_{h,P\pi^\star,\dagger}^{\pi^\star}} \left[ \left[ \phi_i^{V_{h+1,P}^{\pi^\star,\rho}}(s_h,a_h) \right]_{\alpha_h^\star}^2 \left[ \phi_i^{V_{h+1,P}^{\pi^\star,\rho}}(s_h,a_h) \right]_{\alpha_h^\star}^{2,\top} \right] \boldsymbol{\Lambda}_h^{-1} \right)}$$

$$= \sqrt{\rho} \sum_{h=1}^{H} \beta_h \sqrt{\mathrm{Tr}\left( \boldsymbol{\Lambda}_h^{\chi^2}(\alpha_h^\star; d_{h,P\pi^\star,\dagger}^{\pi^\star}, V_{h+1,P}^{\pi^\star}) \boldsymbol{\Lambda}_h^{-1} \right)}$$

$$\leq \sqrt{\rho} \sum_{h=1}^{H} \beta_h \sqrt{\sup_{x \in \mathbb{R}^d} \frac{x^\top \boldsymbol{\Lambda}_h^{\chi^2}(\alpha_h^\star; d_{h,P\pi^\star,\dagger}^{\pi^\star}, V_{h+1,P}^{\pi^\star}) x}{x^\top \boldsymbol{\Lambda}_h^0 x} \, \mathrm{Tr}(\boldsymbol{\Lambda}_h^0 \boldsymbol{\Lambda}_h^{-1})} \tag{C.16}$$

$$\leq \sqrt{\rho} \sum_{h=1}^{H} \beta_h \sqrt{\frac{1}{K} \cdot H^4 \cdot C^{\pi^\star} \cdot \mathrm{Rank}(\boldsymbol{\Lambda}_h^0)} \tag{C.17}$$

$$= \sqrt{\rho} \sum_{h=1}^{H} \beta_h \sqrt{\frac{1}{K} \cdot H^4 \cdot C^{\pi^\star} \cdot d}$$

$$= \frac{c\sqrt{\rho} H^3 d C^{\pi^\star} \log(K/d^2 p_{\min}\zeta)}{\sqrt{K}}, \tag{C.18}$$

where (C.15) holds due to Jensen's inequality, (C.16) holds due to Lemma D.3 and the fact that $\boldsymbol{\Lambda}_h^0 \preceq \boldsymbol{\Lambda}_h$, (C.17) holds due to Assumption 5.11, and and (C.18) holds by the fact $\lambda = d$ and bounding $\beta_h$ as follows

$$\beta_h = \frac{5}{4}\sqrt{\lambda} + \frac{2}{\sqrt{\lambda}}\left( 2\log\frac{H}{\zeta} + d\log\left(4 + \frac{4\lceil 1/p_{\min}\rceil K}{\lambda d}\right) \right) \leq c\sqrt{d}\log\frac{K}{p_{\min}d^2\zeta}.$$

Thus, we have

$$\mathrm{SubOpt}(\hat{\pi}, s_1, \rho)$$

$$\leq \frac{c\sqrt{\rho}H^3 dC^{\pi^\star}\log(K/d^2 p_{\min}\zeta)}{p_{\min}\sqrt{K}} + \frac{cH^2 dC^{\pi^\star}\log(K/d^2 p_{\min}\zeta)}{p_{\min}\sqrt{K}} + \frac{c\sqrt{\rho}H^{5/2} dC^{\pi^\star}\log(K/d^2 p_{\min}\zeta)}{p_{\min}\sqrt{K}}$$

$$\leq \frac{c\left(\sqrt{\rho}H^3 + H^2\right) dC^{\pi^\star}\log(K/d^2 p_{\min}\zeta)}{\sqrt{K}}.$$

We complete the proof of Theorem 5.12. $\qquad\square$

# D  Auxiliary lemmas

**Lemma D.1** (Lemma 1 of [22]). *Let* $\{\mathcal{F}_t\}_{t=0}^{\infty}$ *be a filtration. Let* $\{\boldsymbol{\delta}_t\}_{t=1}^{\infty}$ *be an* $\mathbb{R}^N$-*valued stochastic process such that* $\boldsymbol{\delta}_t$ *is* $\mathcal{F}_t$ *measurable one-hot vector. Furthermore, assume* $\mathbb{E}[\boldsymbol{\delta}_t | \mathcal{F}_{t-1}] = \boldsymbol{p}_t$ *and define* $\boldsymbol{\epsilon}_t = \boldsymbol{p}_t - \boldsymbol{\delta}_t$. *Let* $\{\mathbf{x}_t\}_{t=1}^{\infty}$ *be a sequence of* $\mathbb{R}^{N \times d}$-*valued stochastic process such that* $\mathbf{x}_t$ *is* $\mathcal{F}_{t-1}$ *measurable and* $\|\boldsymbol{x}_{t,i}\|_2 \leq 1, \forall i \in [N]$. *Let* $\{\lambda_t\}_{t=1}^{\infty}$ *be a sequence of non-negative scalars. Define*

$$Y_t = \sum_{i=1}^{t}\sum_{j=1}^{N} \boldsymbol{x}_{i,j}\boldsymbol{x}_{i,j}^\top + \lambda_t \mathbf{I}_d, \qquad S_t = \sum_{i=1}^{t}\sum_{j=1}^{N} \epsilon_{i,j}\boldsymbol{x}_{i,j}.$$

*Then, for any* $\zeta \in (0,1)$, *with probability at least* $1 - \zeta$, *we have for all* $t \geq 1$,

$$\|S_t\|_{Y_t^{-1}} \leq \frac{\sqrt{\lambda_t}}{4} + \frac{4}{\sqrt{\lambda_t}}\log\left(\frac{2^d \det(Y_t)^{1/2}\lambda_t^{-d/2}}{\zeta}\right).$$

**Lemma D.2** (Lemma 2 of [22])**.** *Let* $\zeta \in (0,1)$, *then for any* $k \in [K]$ *and simultaneously for all* $h \in [H]$, *with probability at least* $1 - \delta$, *it holds that*

$$\boldsymbol{\theta}_h^{\star} \in \mathcal{C}_{k,h} \text{ where } \mathcal{C}_{k,h} = \{\|\boldsymbol{\theta} - \boldsymbol{\theta}_{k,h}\|_{\lambda_{k,h}} \le \beta_k\}$$

*with* $\beta_k = (B + \frac{1}{4})\sqrt{\lambda_k} + \frac{2}{\sqrt{\lambda_k}}(2\log(\frac{H}{\zeta}) + d\log(4 + \frac{4SK}{\lambda_k d}))$.

**Lemma D.3** (Lemma 15 of [43])**.** *Suppose* $A_1, A_2, A_3 \in \mathbb{R}^{d \times d}$ *are semipositive definite matrices, then we have*

$$\mathrm{Tr}(A_1 A_2) \le \sigma_{\max}(A_3^{-1/2} A_1 A_3^{-1/2}) \mathrm{Tr}(A_3 A_2),$$

*where*

$$\sigma_{\max}(A_3^{-1/2} A_1 A_3^{-1/2}) = \sup_{x \in \mathbb{R}^d} \frac{x^\top A_1 x}{x^\top A_3 x}.$$

**Lemma D.4.** *(Strong duality for TV [41, Lemma 1]). Given any probability measure* $\mu^0$ *over* $\mathcal{S}$, *a fixed uncertainty level* $\rho$, *the uncertainty set* $\mathcal{U}^\rho(\mu^0) = \{\mu : \mu \in \Delta(\mathcal{S}), D_{TV}(\mu||\mu^0) \le \rho\}$, *and any function* $V : \mathcal{S} \to [0, H]$, *we obtain*

$$
\inf_{\mu \in \mathcal{U}^\rho(\mu^0)} \mathbb{E}_{s \sim \mu} V(s) = \max_{\alpha \in [V_{\min}, V_{\max}]} \Big\{ \mathbb{E}_{s \sim \mu^0}[V(s)]_\alpha
$$
$$
- \rho\big(\alpha - \min_{s'}[V(s')]_\alpha\big)\Big\}, \tag{D.1}
$$

*where* $[V(s)]_\alpha = \min\{V(s), \alpha\}$, $V_{\min} = \min_s V(s)$ *and* $V_{\max} = \max_s V(s)$. *Notably, the range of* $\alpha$ *can be relaxed to* $[0, H]$ *without impacting the optimization.*

**Lemma D.5.** *(Strong duality for KL [17, Theorem]) Suppose* $f(x)$ *has a finite moment generating function in some neighborhood around* $x = 0$, *then for any* $\sigma > 0$ *and a nominal distribution* $P^0$, *we have*

$$
\sup_{P \in \mathcal{U}^\sigma(P^0)} \mathbb{E}_{X \sim P}[f(X)] = \inf_{\lambda \ge 0} \left\{ \lambda \log \mathbb{E}_{X \sim P^0}\left[\exp\left(\frac{f(X)}{\lambda}\right)\right] + \lambda\sigma \right\}.
$$

**Lemma D.6.** *(Strong duality for* $\chi^2$ *[41, Lemma 2]) Consider any probability vector* $P \in \Delta(\mathcal{S})$, *any fixed uncertainty level* $\sigma$, *and the uncertainty set* $\mathcal{U}^\sigma(P) := \mathcal{U}^\sigma_{\chi^2}(P)$. *For any vector* $V \in \mathbb{R}^{\mathcal{S}}$ *obeying* $V \ge 0$, *one has*

$$
\inf_{\mathcal{P} \in \mathcal{U}^\sigma(P)} \mathbb{E}_{s \sim \mathcal{P}} V(s) = \max_{\alpha \in [\min_s V(s), \max_s V(s)]} \left\{ \mathbb{E}_{s \sim P}[V(s)]_\alpha - \sqrt{\sigma \, \mathrm{Var}_P([V]_\alpha)} \right\},
$$

*where* $\mathrm{Var}_P(V) = \mathbb{E}_{s \sim P} V^2(s) - (\mathbb{E}_{s \sim P} V(s))^2$.

# E  Practical algorithms

Despite Algorithm 1 is provably efficient and enables robust policy learning, the construction of the robust policies in (4.3) relies on an optimization oracle. Hence, Algorithm 1 is not computational tractable. In this section, we propose practical algorithms for numerical experiments.

We use value iteration to iteratively estimate the optimal robust Q-function in a backward fashion and take the corresponding greedy policy as the robust policy estimation. Specifically, for any step $h \in [H]$, given an estimated robust value function $\widehat{V}_{h+1}^\rho : \mathcal{S} \to [0, H]$, we dive into the one step robust Bellman operation on $\widehat{V}_{h+1}^\rho$. First, for any $(s, a) \in \mathcal{S} \times \mathcal{A}$, define $\phi^{\widehat{V}_{h+1}^\rho}(s, a) = \int_{\mathcal{S}} \phi(s'|s, a)\widehat{V}_{h+1}^\rho(s')ds'$. One step robust Bellman operation on $\widehat{V}_{h+1}^\rho$ leads to

$$
\begin{aligned}
Q_h^\rho(s, a) &= r_h(s, a) + \inf_{P_h(\cdot|s,a) \in \mathcal{U}_h^\rho(s,a;\boldsymbol{\theta}^0)} \mathbb{E}_{s' \sim P_h(\cdot|s,a)} \widehat{V}_{h+1}^\rho(s') \\
&= r_h(s, a) + \inf_{\boldsymbol{\theta}_h \in \boldsymbol{\Theta}_h} \int_{\mathcal{S}} \langle \phi(s'|s, a), \boldsymbol{\theta}_h \rangle \widehat{V}_{h+1}^\rho(s')ds' \\
&= r_h(s, a) + \inf_{\boldsymbol{\theta}_h \in \boldsymbol{\Theta}_h} \left\langle \int_{\mathcal{S}} \phi(s'|s, a)\widehat{V}_{h+1}^\rho(s')ds', \boldsymbol{\theta}_h \right\rangle
\end{aligned}
$$

$$= r_h(s,a) + \inf_{\boldsymbol{\theta}_h \in \boldsymbol{\Theta}_h} \left\langle \boldsymbol{\phi}^{\widehat{V}^\rho_{h+1}}(s,a), \boldsymbol{\theta}_h \right\rangle$$

$$= r_h(s,a) + \inf_{\boldsymbol{\theta}_h \in \boldsymbol{\Theta}_h} \mathbb{E}_{i \sim \boldsymbol{\theta}_h} \phi_i^{\widehat{V}^\rho_{h+1}}(s,a). \tag{E.1}$$

The infimum term in (E.1) can be solved by optimizing their duality. By the duality formulation in Lemma D.4, for TV-divergence defined $\boldsymbol{\Theta}_h$, we have

$$Q_h^\rho(s,a) = r_h(s,a) + \max_{\alpha \in [0,H]} \left\{ \mathbb{E}_{i \sim \boldsymbol{\theta}_h^0} \left[ \phi_i^{\widehat{V}^\rho_{h+1}}(s,a) \right]_\alpha - \rho \Big( \alpha - \min_i \left[ \phi_i^{\widehat{V}^\rho_{h+1}}(s,a) \right]_\alpha \Big) \right\}.$$

For KL-divergence defined $\boldsymbol{\Theta}_h$, by the duality formulation in Lemma D.5, we have

$$Q_h^\rho(s,a) = r_h(s,a) + \sup_{\lambda \in [0, H/\rho]} \left\{ -\lambda \log \mathbb{E}_{i \sim \boldsymbol{\theta}_h^0} \exp \left\{ -\phi_i^{\widehat{V}^\rho_{h+1}}(s,a)/\lambda \right\} - \lambda\rho \right\}.$$

For $\chi^2$-divergence, by the duality formulation in Lemma D.6, we have

$$Q_h^\rho(s,a) = r_h(s,a) + \max_{\alpha \in [0,H]} \left\{ \mathbb{E}_{i \sim \boldsymbol{\theta}_h^0} \left[ \phi_i^{\widehat{V}^\rho_{h+1}}(s,a) \right]_\alpha - \sqrt{\rho \operatorname{Var}_{i \sim \boldsymbol{\theta}_h^0} \left( \left[ \phi_i^{\widehat{V}^\rho_{h+1}}(s,a) \right]_\alpha \right)} \right\}.$$

Finally, substituting the unknown $\boldsymbol{\theta}_h^0$ in above equations with an estimation $\hat{\boldsymbol{\theta}}_h$, We estimate $\inf_{\boldsymbol{\theta}_h \in \boldsymbol{\Theta}} \mathbb{E}_{i \sim \boldsymbol{\theta}_h} \phi_i^{\widehat{V}^{\rho,k}_{h+1}}(s,a)$ as follows.

$$\text{TV: } \widehat{\inf_{\boldsymbol{\theta}_h \in \boldsymbol{\Theta}_h}} \mathbb{E}_{i \sim \boldsymbol{\theta}_h} \phi_i^{\widehat{V}^{\rho,k}_{h+1}}(s,a) = \max_{\alpha \in [0,H]} \left\{ \mathbb{E}_{i \sim \hat{\boldsymbol{\theta}}_h} \left[ \phi_i^{\widehat{V}^\rho_{h+1}}(s,a) \right]_\alpha - \rho \Big( \alpha - \min_i \left[ \phi_i^{\widehat{V}^\rho_{h+1}}(s,a) \right]_\alpha \Big) \right\} \tag{E.2}$$

$$\text{KL: } \widehat{\inf_{\boldsymbol{\theta}_h \in \boldsymbol{\Theta}_h}} \mathbb{E}_{i \sim \boldsymbol{\theta}_h} \phi_i^{\widehat{V}^{\rho,k}_{h+1}}(s,a) = \sup_{\lambda \in [0, H/\rho]} \left\{ -\lambda \log \mathbb{E}_{i \sim \hat{\boldsymbol{\theta}}_h} \exp \left\{ -\phi_i^{\widehat{V}^\rho_{h+1}}(s,a)/\lambda \right\} - \lambda\rho \right\} \tag{E.3}$$

$$\chi^2\text{: } \widehat{\inf_{\boldsymbol{\theta}_h \in \boldsymbol{\Theta}_h}} \mathbb{E}_{i \sim \boldsymbol{\theta}_h} \phi_i^{\widehat{V}^{\rho,k}_{h+1}}(s,a) = \max_{\alpha \in [0,H]} \left\{ \mathbb{E}_{i \sim \hat{\boldsymbol{\theta}}_h} \left[ \phi_i^{\widehat{V}^\rho_{h+1}}(s,a) \right]_\alpha \sqrt{\rho \operatorname{Var}_{i \sim \hat{\boldsymbol{\theta}}_h} \left( \left[ \phi_i^{\widehat{V}^\rho_{h+1}}(s,a) \right]_\alpha \right)} \right\}, \tag{E.4}$$

where $\hat{\boldsymbol{\theta}}_h$ is an estimation of $\boldsymbol{\theta}_h^0$. In implementation, the optimization in (E.2) - (E.4) can be approximately solved by the heuristic Nelder-Mead method [31]. Finally, we propose two approaches to estimate the unknown parameter $\boldsymbol{\theta}_h^0$ from the offline dataset.

**Transition-targeted regression** Follow the transition-targeted regression proposed in (4.1), we get the estimation of $\boldsymbol{\theta}_h^0$, $\hat{\boldsymbol{\theta}}_h$. Note that the parameter estimation procedure of the transition-targeted regression only depends on the transition information in the offline dataset. Thus, it can be conducted simultaneously for all stages $h \in [H]$ and obtain $[\hat{\boldsymbol{\theta}}_h]_{h \in [H]}$ all at once.

**Value-targeted regression** As an alternative choice for parameter estimation, the value targeted regression is an approach well studied in theory studies on standard RL [19, 2, 58]. To illustrate its application in robust RL, we also proposed an algorithm to learn robust policies based on the value targeted regression for parameter $\{\boldsymbol{\theta}_h^0\}_{h=1}^H$ estimation. In particular, at stage $h$ with the estimated robust value function $\widehat{V}^\rho_{h+1}$, we estimate $\boldsymbol{\theta}_h^0$ via the following ridge regression:

$$\operatorname*{argmin}_{\boldsymbol{\theta} \in \mathbb{R}^d} \sum_{k=1}^K \left[ \left\langle \boldsymbol{\phi}^{\widehat{V}^\rho_{h+1}}(s_h^k, a_h^k), \boldsymbol{\theta} \right\rangle - \widehat{V}^\rho_{h+1}(s_{h+1}^k) \right]^2 + \lambda \|\boldsymbol{\theta}\|_2^2 = \boldsymbol{\Lambda}_h^{-1} \sum_{k=1}^K \boldsymbol{\phi}^{\widehat{V}^\rho_{h+1}}(s_h^k, a_h^k) \widehat{V}^\rho_{h+1}(s_{h+1}^k), \tag{E.5}$$

where $\boldsymbol{\Lambda}_h = \sum_{k=1}^K \boldsymbol{\phi}^{\widehat{V}^\rho_{h+1}}(s_h^k, a_h^k) \left( \boldsymbol{\phi}^{\widehat{V}^\rho_{h+1}}(s_h^k, a_h^k) \right)^\top + \lambda \mathbf{I}$. Note that the parameter estimation procedure of the value-targeted regression is conducted iteratively with the value function estimation. For example, the parameter estimation at stage $h$ depends on the estimated value function at the next stage $h+1$, which in turns depends on the parameter estimation at stage $h+1$. This is different from the parameter estimation procedure of the transition-targeted regression.

Algorithms based on the transition targeted regression (termed as the DRTTR) and the value targeted regression (termed as the DRVTR) are presented in Algorithm 2.

**Algorithm 2** Distributionally Robust Transition (Value) Targeted Regression (DRTTR and DRVTR)

**Require:** Regularization parameter $\lambda$, offline dataset $\mathcal{D}$, robust level $\rho$, initialization $\widehat{V}_{H+1}(\cdot) = 0$.
1: **for** $h = H, \cdots, 1$ **do**
2:     For DRTTR, estimate $\hat{\boldsymbol{\theta}}_h$ by (4.1); For DRVTR, estimate $\hat{\boldsymbol{\theta}}_h$ by (E.5).
3:     Estimate $\widehat{Q}_h^\rho(\cdot, \cdot)$ using (E.2), (E.3) and (E.4) for TV-, KL- and $\chi^2$- divergences, respectively.
4:     $\pi_h(\cdot) \leftarrow \operatorname{argmax}_{a \in \mathcal{A}} \widehat{Q}_h^\rho(\cdot, a), \widehat{V}_h^\rho(\cdot) \leftarrow \max_{a \in \mathcal{A}} \widehat{Q}_h^{k,\rho}(\cdot, a)$
5: **end for**

# F   Simulation study

We now conduct experiments in a simulated environment to show the robustness of policies learned by our algorithms.

## F.1   Experiment setup

We adapt the simulated linear MDP instance proposed by [24] to a linear mixture DRMDP. Note that we have access to difference information in those settings. In particular, for linear DRMDPs, an agent has access to the feature mapping $\boldsymbol{\psi}: (s, a) \to \mathbb{R}^d$. While for linear mixture DRMDPs, an agent has access to basis modes $\boldsymbol{\phi}: (s, a) \to \Delta(\mathcal{S})^d$. Thus, though the DRMDP environments actually remain the same, our implemented algorithms are essentially different from that in [24] due to the different prior information available.

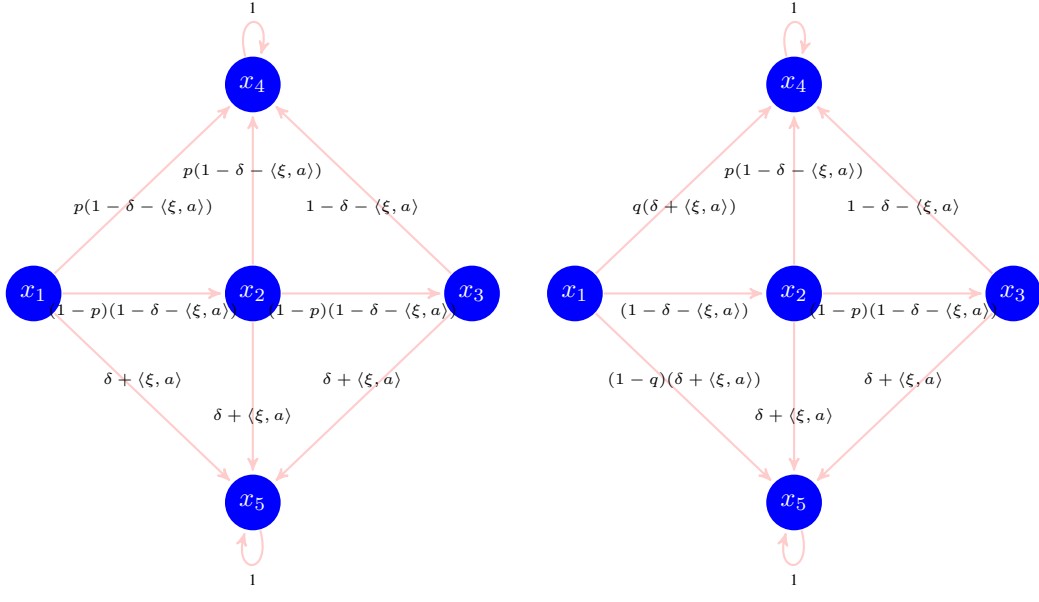

(a) The source environment.                 (b) The target environment.

Figure 2: The source and the target linear MDP environments. The value on each arrow represents the transition probability. For the source MDP, there are five states and three steps, with the initial state being $x_1$, the fail state being $x_4$, and $x_5$ being an absorbing state with reward 1. The target MDP on the right is obtained by perturbing the transition probability at the first step of the source MDP, with others remaining the same.

We present the source domain linear mixture MDP in Figure 2a. Specifically, there are five states $\mathcal{S} = \{x_1, x_2, x_3, x_4, x_5\}$, 16 actions $\mathcal{A} = \{-1, 1\}^4$, and $H = 3$ stages. The basis modes $\boldsymbol{\phi}(\cdot | \cdot, \cdot)$ are

probability measures defined on $\mathcal{S}/x_1$ as follows:

$$\boldsymbol{\phi}(\cdot|x_1,a) = \begin{bmatrix} \phi_1(\cdot|x_1,a) \\ \phi_2(\cdot|x_1,a) \\ \phi_3(\cdot|x_1,a) \end{bmatrix} = \begin{bmatrix} 1-\delta-\langle\boldsymbol{\xi},a\rangle & 0 & \delta+\langle\boldsymbol{\xi},a\rangle & 0 \\ 1-\delta-\langle\boldsymbol{\xi},a\rangle & 0 & 0 & \delta+\langle\boldsymbol{\xi},a\rangle \\ 0 & 0 & 1-\delta-\langle\boldsymbol{\xi},a\rangle & \delta+\langle\boldsymbol{\xi},a\rangle \end{bmatrix},$$

$$\boldsymbol{\phi}(\cdot|x_2,a) = \begin{bmatrix} \phi_1(\cdot|x_2,a) \\ \phi_2(\cdot|x_2,a) \\ \phi_3(\cdot|x_2,a) \end{bmatrix} = \begin{bmatrix} 0 & 1-\delta-\langle\boldsymbol{\xi},a\rangle & \delta+\langle\boldsymbol{\xi},a\rangle & 0 \\ 0 & 1-\delta-\langle\boldsymbol{\xi},a\rangle & 0 & \delta+\langle\boldsymbol{\xi},a\rangle \\ 0 & 0 & 1-\delta-\langle\boldsymbol{\xi},a\rangle & \delta+\langle\boldsymbol{\xi},a\rangle \end{bmatrix},$$

$$\boldsymbol{\phi}(\cdot|x_4,a) = \begin{bmatrix} \phi_1(\cdot|x_4,a) \\ \phi_2(\cdot|x_4,a) \\ \phi_3(\cdot|x_4,a) \end{bmatrix} = \begin{bmatrix} 0 & 0 & 1 & 0 \\ 0 & 0 & 1 & 0 \\ 0 & 0 & 1 & 0 \end{bmatrix},$$

$$\boldsymbol{\phi}(\cdot|x_5,a) = \begin{bmatrix} \phi_1(\cdot|x_5,a) \\ \phi_2(\cdot|x_5,a) \\ \phi_3(\cdot|x_5,a) \end{bmatrix} = \begin{bmatrix} 0 & 0 & 0 & 1 \\ 0 & 0 & 0 & 1 \\ 0 & 0 & 0 & 1 \end{bmatrix}, \tag{F.1}$$

where $\delta$ and $\boldsymbol{\xi}$ are hyperparameters. Apparently, the dimension $d$ equals to three in this instance. It is trivial to verify that the source transition kernels can be represented as $P_h^0(\cdot|\cdot,\cdot) = \boldsymbol{\phi}(\cdot|\cdot,\cdot)^\top \boldsymbol{\theta}_h^0$. The reward functions $r_h(s,a)$ are designed as

$$r_h(s,a) = \boldsymbol{\psi}(s,a)^\top \boldsymbol{\nu}_h, \ \forall (h,s,a) \in [H] \times \mathcal{S} \times \mathcal{A},$$

where

$$\boldsymbol{\psi}(x_1,a) = (1-\delta-\langle\xi,a\rangle, 0, 0, \delta+\langle\xi,a\rangle)^\top,$$
$$\boldsymbol{\psi}(x_2,a) = (0, 1-\delta-\langle\xi,a\rangle, 0, \delta+\langle\xi,a\rangle)^\top,$$
$$\boldsymbol{\psi}(x_3,a) = (0, 0, 1-\delta-\langle\xi,a\rangle, \delta+\langle\xi,a\rangle)^\top,$$
$$\boldsymbol{\psi}(x_4,a) = (0, 0, 1, 0)^\top,$$
$$\boldsymbol{\psi}(x_5,a) = (0, 0, 0, 1)^\top,$$

and

$$\boldsymbol{\nu}_1 = (0,0,0,0)^\top, \ \boldsymbol{\nu}_2 = (0,0,0,1)^\top \ \text{and} \ \boldsymbol{\nu}_3 = (0,0,0,1)^\top.$$

We construct target domains by perturbing the weighting parameter $\boldsymbol{\theta}_1^0$ in the first stage of the source MDP. Specifically, we set $\boldsymbol{\theta}_1 = (q, 1-q, 0)^\top$ where $q$ is a factor that controls the perturbation level. The perturbed target environment is shown in Figure 2b.

**Implementation**  For the offline dataset collection, we simply use the random policy that chooses actions uniformly at random at any $(s,a,h) \in \mathcal{S} \times \mathcal{A} \times [H]$ as the behavior policy $\pi^b$ to collect the offline dataset $\mathcal{D}$. The offline dataset $\mathcal{D}$ contains 500 trajectories collected by the behavior policy $\pi^b$ from the source environment $P^0$. For the setting details of the source environment $P^0$, we set hyperparameters in the defining the source and target environments as $\boldsymbol{\xi} = (1/\|\boldsymbol{\xi}\|_1, 1/\|\boldsymbol{\xi}\|_1, 1/\|\boldsymbol{\xi}\|_1, 1/\|\boldsymbol{\xi}\|_1)^\top$, $\|\boldsymbol{\xi}\|_1 = 0.4$, $p = 0.1$, $\delta = 0.4$, and $q \in [0,1]$. We implement Algorithm 2 with TV, KL and $\chi^2$ divergences on the collected offline dataset $\mathcal{D}$, and denote them as DRTTR-TV, DRTTR-KL DRTTR-chi2, DRVTR-TV, DRVTR-KL and DRVTR-chi2, respectively. Denoting the robust levels of the TV, KL and $\chi^2$ uncertainty set as $\rho_{\text{TV}}, \rho_{\text{KL}}, \rho_{\chi^2}$, we consider two sets of robust levels: $(\rho_{\text{TV}}, \rho_{\text{KL}}, \rho_{\chi^2}) \in \{(0.35, 5, 10), (0.7, 10, 20)\}$. We compare DRTTR and DRVTR with the non-robust algorithms, dubbed as the TTR and VTR respectively, which basically set the robust level $\rho = 0$ in DRTTR and DRVTR. To show the robustness of the learned policies by DRTTR and DRVTR, we test the learned policies on various target environments with different levels of perturbation. We use the 'average reward', which is defined as the averaged cumulative reward among 100 episodes, as a criterion to evaluate performances of robust policies under different target environments. All experiment results are based on 10 replications, and were conducted on a MacBook Pro with a 2.6 GHz 6-Core Intel CPU.

## F.2  Experiment results

Simulation results Algorithm 2 are shown in Figure 3. As the perturbation level increases, the performances of policies learned by the non-robust algorithms TTR and VTR drop drastically when the magnitude of perturbation increases. When the perturbation is large, all robust policies outperform the non-robust policy. This illustrates the robustness of our proposed algorithms. We also conduct additional ablation studies to test the performances of Algorithm 2 in various environments.

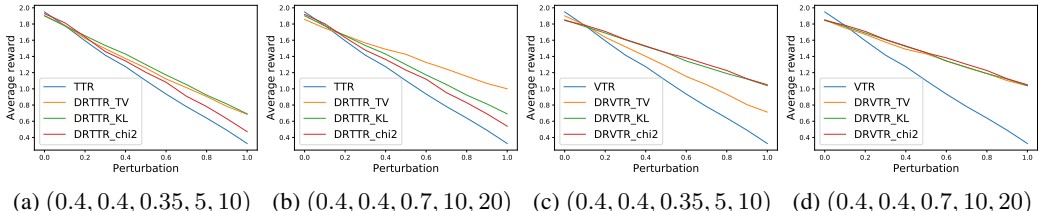

(a) $(0.4, 0.4, 0.35, 5, 10)$    (b) $(0.4, 0.4, 0.7, 10, 20)$    (c) $(0.4, 0.4, 0.35, 5, 10)$    (d) $(0.4, 0.4, 0.7, 10, 20)$

Figure 3: Simulation results of Algorithm 2. Policies are learned from the nominal environment featuring $\boldsymbol{\theta}_1 = (0.1, 0.8, 0.1)$. Numbers in parenthesis represent $(\delta, \|\xi\|_1, \rho_{\text{TV}}, \rho_{\text{KL}}, \rho_{\chi^2})$, respectively. The $x$-axis represents the perturbation level corresponding to different target environments. $\rho_{\text{TV}}, \rho_{\text{KL}}$ and $\rho_{\chi^2}$ are the input uncertainty levels for our DRTTR algorithm.

**Ablation study** We set the weighting parameters in the source environment as

$$\boldsymbol{\theta}_1^0 = \boldsymbol{\theta}_2^0 = (0, 1-p, p)^\top, \tag{F.2}$$

where $p = 0.1$ is the hyperparameter that controls the mixture weights. We set the hyperparameter $\boldsymbol{\xi} = (1/\|\boldsymbol{\xi}\|_1, 1/\|\boldsymbol{\xi}\|_1, 1/\|\boldsymbol{\xi}\|_1, 1/\|\boldsymbol{\xi}\|_1)^\top$ and consider different choices of $\|\boldsymbol{\xi}\|_1 \in \{0.3, 0.4\}$. We consider difference levels of hyperparameter $\delta \in \{0.3, 0.4\}$. We implement DRTTR with TV, KL and $\chi^2$ divergence, as well as DRVTR with TV, KL and $\chi^2$ divergence on the collected offline dataset $\mathcal{D}$, and denote them as DRTTR-TV, DRTTR-KL DRTTR-chi2, DRVTR-TV, DRVTR-KL and DRVTR-chi2, respectively. Denoting the robust levels of the TV, KL and $\chi^2$ uncertainty set as $\rho_{\text{TV}}, \rho_{\text{KL}}, \rho_{\chi^2}$, we consider two sets of robust levels: $(\rho_{\text{TV}}, \rho_{\text{KL}}, \rho_{\chi^2}) \in \{(0.35, 5, 10), (0.7, 10, 20)\}$. We test the learned polices in testing environments with $q \in [0, 1]$.

Experiment results are shown in Figure 4 and Figure 5. Moreover, we also consider the source domain with the weighting parameter at the first stage $\boldsymbol{\theta}_1^0 = (p, 1-2p, p)^\top$, while all other parameters remain the same. Experiment results are shown in Figure 6 and Figure 7. We can conclude that in most cases Algorithm 2 can learn robust policies compared to their non-robust counterparts. In particular, when the perturbation becomes larger, all robust policies outperform non-robust policies in terms of the cumulative reward.

As for the choice of uncertainty set, it requires certain prior knowledge of the nominal dynamics and distribution shift. This actually is a long-standing problem in this field that limited theoretical guarantees are available to guide the choice of uncertainty sets. In practice, one can learn policies corresponding to different divergence defined uncertainty sets and then compare their performances in testing environments.

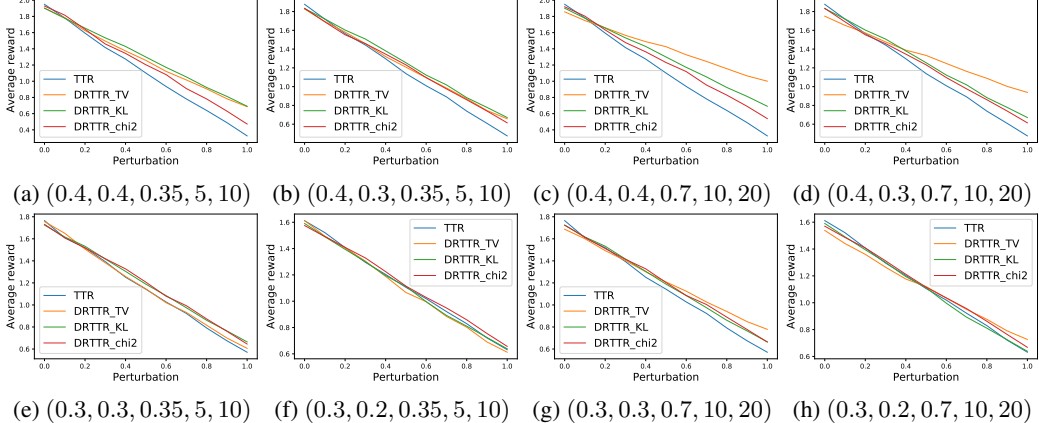

(a) $(0.4, 0.4, 0.35, 5, 10)$   (b) $(0.4, 0.3, 0.35, 5, 10)$   (c) $(0.4, 0.4, 0.7, 10, 20)$   (d) $(0.4, 0.3, 0.7, 10, 20)$

(e) $(0.3, 0.3, 0.35, 5, 10)$   (f) $(0.3, 0.2, 0.35, 5, 10)$   (g) $(0.3, 0.3, 0.7, 10, 20)$   (h) $(0.3, 0.2, 0.7, 10, 20)$

Figure 4: Simulation results of DRTTR under different source domains. Policies are learned from the nominal environment featuring $\boldsymbol{\theta}_1 = (0, 0.9, 0.1)$. Numbers in parenthesis represent $(\delta, \|\xi\|_1, \rho_{\text{TV}}, \rho_{\text{KL}}, \rho_{\chi^2})$, respectively. The $x$-axis represents the perturbation level corresponding to different target environments. $\rho_{\text{TV}}, \rho_{\text{KL}}$ and $\rho_{\chi^2}$ are the input uncertainty levels for our DRTTR algorithm.

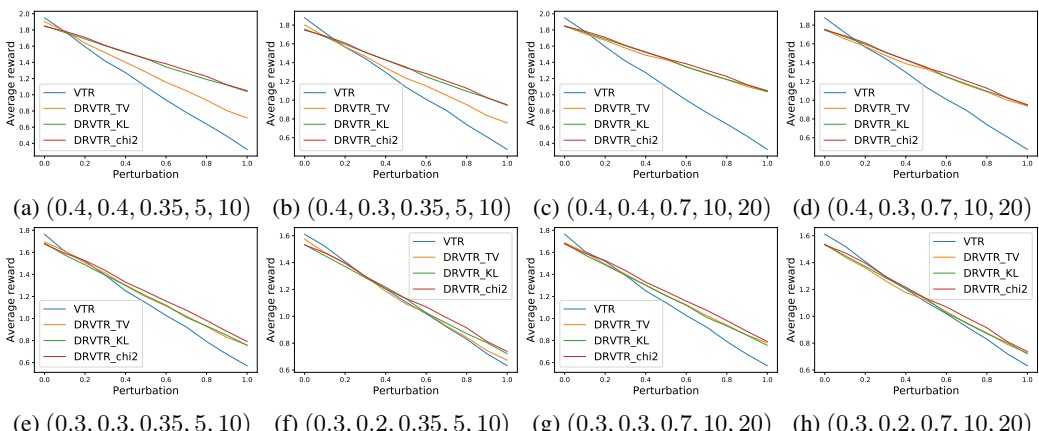

Figure 5: Simulation results of DRVTR under different source domains. Policies are learned from the nominal environment featuring $\boldsymbol{\theta}_1 = (0, 0.9, 0.1)$. The $x$-axis represents the perturbation level corresponding to different target environments. $\rho_{\mathrm{TV}}, \rho_{\mathrm{KL}}$ and $\rho_{\chi^2}$ are the input uncertainty levels for our DRVTR algorithm.

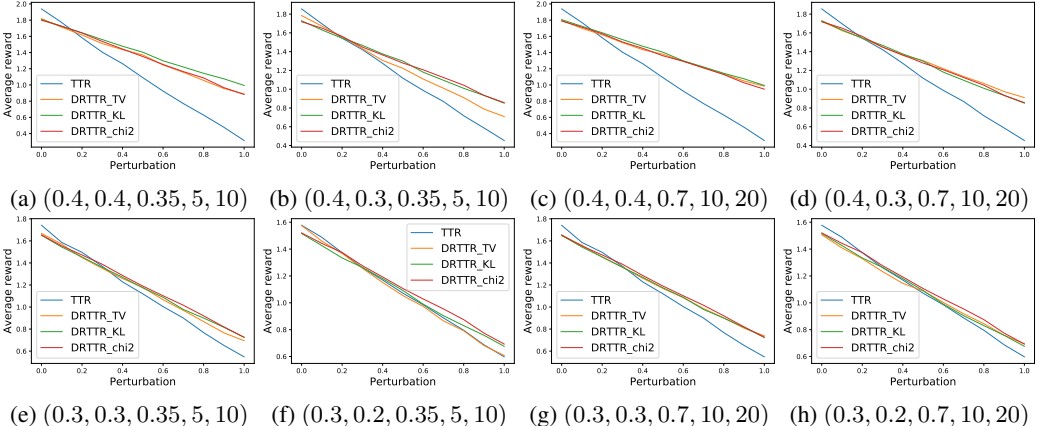

Figure 6: Simulation results of DRTTR under different source domains. Policies are learned from the nominal environment featuring $\boldsymbol{\theta}_1 = (0.1, 0.8, 0.1)$. Numbers in parenthesis represent $(\delta, \|\xi\|_1, \rho_{\mathrm{TV}}, \rho_{\mathrm{KL}}, \rho_{\chi^2})$, respectively. The $x$-axis represents the perturbation level corresponding to different target environments. $\rho_{\mathrm{TV}}, \rho_{\mathrm{KL}}$ and $\rho_{\chi^2}$ are the input uncertainty levels for our DRTTR algorithm.

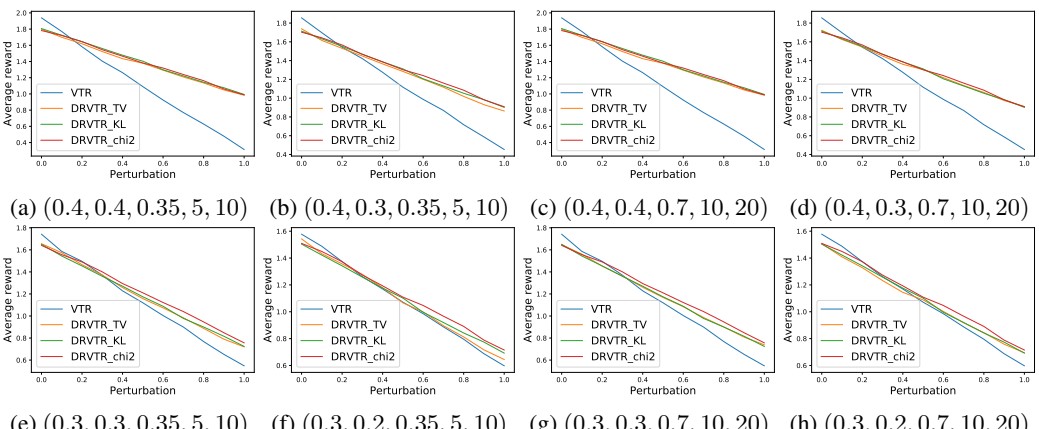

Figure 7: Simulation results of DRVTR under different source domains. Policies are learned from the nominal environment featuring $\boldsymbol{\theta}_1 = (0.1, 0.8, 0.1)$. Numbers in parenthesis represent $(\delta, \|\xi\|_1, \rho_{\text{TV}}, \rho_{\text{KL}}, \rho_{\chi^2})$, respectively. The $x$-axis represents the perturbation level corresponding to different target environments. $\rho_{\text{TV}}, \rho_{\text{KL}}$ and $\rho_{\chi^2}$ are the input uncertainty levels for our DRVTR algorithm.

