# OpenReview forum: "Linear Mixture Distributionally Robust Markov Decision Processes"
_NeurIPS.cc/2025/Conference — NeurIPS 2025 poster_

### Official Review · Reviewer_okkx · 2025-06-30

**Clarity:** 3
**Significance:** 2
**Originality:** 3
**Rating:** 4
**Confidence:** 2

**Summary:**

In this paper, the author proposes a linear mixture distributionally robust Markov decision process (DRMDP) framework, which differs from the more common setting where the ambiguity set is defined as a ball centered around a nominal transition kernel. The author also provides proofs that the dynamic programming principles hold for linear mixture DRMDPs. This result serves as a solid theoretical foundation for future research. In datails,  the author analyzes the case of $\phi$-divergence-based ambiguity sets and presents an error bound analysis.

**Questions:**

1. As mentioned in the weaknesses, the motivation for using the linear mixture ambiguity set is not entirely clear. It would be helpful if the author could explain under what conditions or in which scenarios this type of ambiguity set is preferable to more common settings,. Are there specific applications where the linear mixture ambiguity set offers clear advantages? A more detailed discussion or empirical comparison would strengthen the motivation for this approach.

2. Since the linear mixture ambiguity set is formed by combining several latent base models in a linear way, it is unclear how well the method performs when the true distribution is non-convex or multimodal. For example, if the underlying distribution contains multiple distinct modes, the linear mixture structure may not be flexible enough to capture such complexity. Can the author extend the method to better handle multimodal distributions?

3. In the experiments, the author compares the non-robust method VTR with the DRTTR-based method. Can we have a comparison between the proposed distributionally robust method using the linear mixture ambiguity set and a method based on an $(s,a)$-rectangular ambiguity set?

**Ethical Concerns:**

["NO or VERY MINOR ethics concerns only"]

**Final Justification:**

Keep score

**Limitations:**

Yes

**Quality:**

3

**Strengths And Weaknesses:**

Strengths: The structure of the paper is clear, and the author uses several figures to illustrate the differences between the linear mixture ambiguity set and the more common settings. These visualizations help the reader better understand the key ideas and contributions of the work.

Weaknesses: Although the author provides strong theoretical results, the analysis relies on a number of assumptions. This raises the question of whether real-world applications can satisfy all these assumptions, especially since each theorem introduces new conditions. While the theoretical contributions are well-supported, the performance evaluation is based on training with an offline dataset. A demonstration using a real-world application would be more convincing and could better highlight the practical effectiveness of the proposed approach. Moreover, common ambiguity sets, such as those based on Wasserstein distance or $\phi$-divergence, have been shown to perform well in many practical settings. Therefore, it would be helpful if the author could provide more detailed guidance on when to choose the linear mixture ambiguity set over these existing alternatives.

---

> ### Author Rebuttal · Authors · 2025-07-31
>
> We thank the reviewer for their valuable time and effort in providing detailed feedback on our work. We hope our response will fully address all of the reviewer's points.
>
> ---
> > **Q1:** Practical motivation for linear mixture DRMDPs. When should we choose linear mixture DRMDPs over other frameworks?
>
> **A1:** Thanks for the interesting question. To answer this question, we would like to lay out several practical motivations for our framework. Mixture models, in particular Gaussian Mixture Models (GMMs), are widely studied in fields of optimal control and RL. For example:
>
> **Model Predictive Control (MPC)**: [1] studies the density steering for discrete-time linear systems, a variant of optimal mass transport problem widely used in applications such as controlling a swarm of robots/drones to achieve a desired spatial distribution. They model the stochastic dynamical system using GMMs and derive its optimal control policy. [2] studies a stochastic MPC algorithm for linear systems subject to additive Gaussian mixture disturbances. They consider a vehicle control case study in which the vehicle must maintain its position on an ill-maintained road.
>
> **Meta RL**: Meta learning is one way to increase the data efficiency of learning algorithms by generalizing learned concepts from a set of training tasks to unseen, but related, tasks. Recent works on Meta RL model the dynamics of a new system as a mixture of expert systems, thus realizing knowledge transfer. In particular, [3,4] model the dynamics of the expert systems using Gaussian process with latent embedding. And the dynamics of the new system is constructed by a mixture of Gaussian processes with the distribution over the latent embedding being the mixture weights, which should be learned from data. In experiments, their meta-learning models effectively generalizes to novel tasks under various environments such as Cart-pole & Double-pendulum swing-up, HalfCheetah and Highway-Intersection. Our framework could potentially enhance robustness in unseen tasks.
>
> **World Models for model-based RL**: Model-based Deep Reinforcement Learning (RL) assumes the availability of a model of an environment’s underlying transition dynamics, which are stochastic in nature and oftentimes multimodal. [5,6] use Mixture-density Networks to construct World Models to capture the multimodality in dynamics. Then model-based RL methods are implemented based on the constructed World Models to solve various tasks such as car racing and Inverse Sine Wave.
>
> The basis modes in our work can be the Gaussian components in GMMs, dynamics of expert systems, or mixture components in the Mixture-density Networks. One potential application of our framework would be the meta-RL. For example, assume we have constructed expert system dynamics (basis modes) from huge amounts of historical data, and would like to solve a new task whose dynamics is a mixture distribution of expert system dynamics. With limited data collected from the new task, there is high uncertainty in weighting parameter estimation. In such cases, the linear mixture DRMDP framework leverages the latent structure in dynamics of the new task, and thus is conceptually more favorable than other DRMDP framework. Other applications involve robust control of drones or vehicles to hedge against environmental uncertainties and distributional perturbations, and robust decision making in model-based RL by modeling the uncertainty in the World Model (specifically, in the weights).
> Moreover, it serves as an interesting future research direction to also incorporate the uncertainty in basis modes into consideration for robust decision making.
>
>
> ---
> > **Q2:** Linear mixture structure for multimodal distributions.
>
>
>
> **A2:** Thank you for the interesting question. We highlight that the linear mixture structure can naturally model multimodal transitions. In particular, each basis mode can represent a component distribution with corresponding local maximum in the probability density function or probability mass function, such as gaussian distributions with different means and variances. And the mixture of basis modes capture the multimodality in the dynamics, say, for a given state-action pair, the next state follows multiple distinct patterns. For example, [5] discuss the multimodality in dynamics of many real-world RL applications. They introduce world models using mixture distribution networks to approximate the multimodal dynamics, which could potentially be adapted to our framework to construct the basis modes.
>
> ---
> > **Q3:** Comparison between the proposed distributionally robust method using the linear mixture ambiguity set and a method based on an (s,a)-rectangular ambiguity set.
>
> **A3:** We can justify this point. In this work we mainly focus on establishing the framework, which is shown in Section 2 to have advantages in modelling the uncertainty in cases where existing frameworks fall short, from scratch and designing algorithms that achieve provable efficiency. The simple simulation environment is just to validate our theoretical results, i.e., to showcase the robustness of our proposed algorithms. We also would like to note that the linear mixture DRMDP assumes prior information, i.e. the access to basic modes, for algorithm design. It’s not fair to directly compare algorithms designed for different DRMDP settings because they either require other prior information or do not require prior information at all. Thus, we instead compare those two algorithms designed for the linear mixture DRMDP.
>
> As an interesting future work, we would like to explore when it is appropriate to model the change in the dynamics as a linear mixture uncertainty set and how to construct the latent modes in complex environments like MoJuCo. In those real-world applications, we can have a fair comparison between different DRMDP frameworks. However, this requires considerable effort, which is beyond the scope of this work.
>
> ---
> **References:**
>
> [1] Balci et. al., 2024, Density steering of gaussian mixture models for discrete-time linear systems
>
> [2] Engelaar et. al., 2024, Stochastic MPC for Finite Gaussian Mixture Disturbances with Guarantees
>
> [3] Sæmundsson et. al., 2018, Meta reinforcement learning with latent variable gaussian processes
>
> [4] Xu et. al., 2020, Task-agnostic online reinforcement learning with an infinite mixture of gaussian processes
>
> [5] Sedlmeier et. al., 2021, Quantifying multimodality in world models
>
> [6] Ha et. al., 2018, Recurrent world models facilitate policy evolution
>
> ---
> We hope we have addressed all of your questions/concerns. If you have any further questions, we would be more than happy to answer them and if you don’t, would you kindly consider increasing your score?

---

> > ### Comment · Reviewer_okkx · 2025-08-05
> >
> > Thanks so much for the response. I’m satisfied with the explanation and I’ll maintain the score.

---

### Official Review · Reviewer_sAJt · 2025-07-02

**Clarity:** 2
**Significance:** 3
**Originality:** 2
**Rating:** 5
**Confidence:** 3

**Summary:**

The authors tackle the problem of solving Markov decision process problems with uncertainty in the transition matrices.
They introduce the notion of distributionally robust linear mixture MDPs, where the transition matrices are assumed to be expressed as linear combinations of known "basis latent modes". The weights used for this combination are assumed to lie in an uncertainty set.
They consider various f-divergence-based ambiguity sets and develop a meta-algorithm to solve the problem.
They provide theoretical guarantees on the suboptimality of the policy obtained using this algorithm.
Finally, in the appendix, they provide some numerical results demonstrating their approach and the benefit of using a distributionally robust policy under increasing noise.

**Questions:**

Questions
1. Can you discuss the dependence of suboptimality on the size of the uncertainty set?
3. Can you discuss the dependence of your models on the number of data trajectories available for training? DRO approaches are expected to be particularly useful if limited data is available. Does that appear true in your setting?
4. Could you discuss the impact of the robustness approach on scalability, especially compared to the standard (s, a) and d-uncertainty sets?
5. Can the authors also discuss a bit about how difficult or easy it is to obtain good estimates of the latent modes, which are combined to obtain the transition matrices?

**Ethical Concerns:**

["NO or VERY MINOR ethics concerns only"]

**Final Justification:**

The authors have responded to my questions in detail. As I had already recommended the acceptance of the paper, I will maintain my score.

**Limitations:**

Limitations
The work is primarily theoretical, so there is not much to address in terms of negative societal impact.
I would like it if the authors discuss the potential limitations of their choice to use mixture models. What kinds of worst cases can they address or not, and so on.

**Paper Formatting Concerns:**

It looks to be fine.

**Quality:**

3

**Strengths And Weaknesses:**

Strengths
1. The idea behind the paper makes sense and is somewhat original.
2. The authors provide performance guarantees for their estimated solution
3. They also theoretically compare their uncertainty models to other uncertainty sets.

Weaknesses
1. The notation presented in the paper is quite challenging to read.
2. The example provided in the appendix involves very few states.
3. Lack of numerical comparison to standard (s, a) or d-uncertainty sets, which makes it difficult to evaluate the presented uncertainty models.

---

> ### Author Rebuttal · Authors · 2025-07-31
>
> We thank the reviewer for their valuable time and effort in providing detailed feedback on our work. We hope our response will fully address all of the reviewer's points.
>
> ---
> > **Q1:** Dependence of suboptimality on the size of the uncertainty set.
>
> **A1:** Thank you for your question. For the case of TV divergence, the suboptimality bound does not depend on the radius $\rho$, as shown in Theorem 4.6 of our submission. For the case of KL divergence, the suboptimality bound is inversely proportional to the radius $\rho$ as shown in Theorem 4.11 of our submission. This dependence aligns with prior works on DRMDP with KL divergence defined uncertainty set (see Theorem 3 in [1] and Corollary 4.1 in [2]), arising from the geometry of KL divergence in analysis. For $\chi^2$ divergence, the suboptimality bound is proportional to $(\sqrt{\rho}H^3+H^2)$ as shown in Theorem 4.13 of our submission. For $\rho=O(1/H^2)$, this term is in order of $O(H^2)$ and our bound matches that for non-robust linear mixture MDP; for moderate $\rho$, our bound suggests that policy learning would be harder due to the complex geometry of the $\chi^2$ divergence uncertainty set. A similar pattern has also been identified by a prior work [3] on tabular DRMDPs with $\chi^2$ divergence defined uncertainty sets.
>
> ---
> > **Q2:** Dependence of your models on the number of data trajectories available for training.
>
> **A2:** Our suboptimality bound can be translated into sample complexity bounds. To learn an $\epsilon$-optimal policy, we need sample sizes in order of $\tilde{O}(\frac{d^2H^4(C^{\pi^{\star}})^2}{\epsilon^2})$ for TV divergence, $\tilde{O}(\frac{d^2H^4(C^{\pi^{\star}})^2e^{2H/{\underline{\lambda}}}}{\rho^2\epsilon^2})$ for KL divergence, and $\tilde{O}(\frac{d^2(\sqrt{\rho}H^3+H^2)^2(C^{\pi^{\star}})^2}{\epsilon^2})$ for $\chi^2$ divergence.
>
> Classic DRO approaches are expected to be effective in scenarios with limited data because they are explicitly designed to guard against **uncertainty from insufficient data samples**. In contrast, our current framework mostly focuses on uncertainty from dynamics shift, which is also common in real world applications.
> Moreover, our DRMDP framework hedging against dynamics shifts can also be adapted to achieve robustness w.r.t. parameter estimation uncertainty due to limited data. Actually this idea has been explored in the prior work [4], where they show DRMDP can be used to solve offline RL. Experiments therein demonstrate DRO approaches achieve significantly better performance than non-robust approaches with limited samples. Thus, we can expect the same is true for our framework. We leave a comprehensive examination for future study.
>
> ---
> > **Q3:** The impact of the robustness approach on scalability, especially compared to the standard (s, a) and d-uncertainty sets?
>
> **A3:** From a theoretical perspective, our suboptimality bound does not depend on the size of state-action space. Thus even when the state-action space is of high dimensional and infinitely large, our approach scales well thanks to the linear mixture structure. d-rectangular DRMDPs also can scale, leveraging a different modeling on latent structure of dynamics and uncertainty set. The linear mixture MDP or linear MDP assumption on the nominal transition kernel naturally admits linear representation in terms of the dynamics or Q-function, thus enabling linear function approximation for scalability. While for $(s,a)$-rectangular DRMDPs, most existing model-based and model-free approaches [1,3,6] cannot scale, they either rely on frequency-based transition estimation or rely on a Q-table for robust Bellman update. Existing literature has also studied general function approximation for scalability in the context of $(s,a)$-rectangular DRMDPs. For example, [5] study $(s,a)$-rectangular DRMDPs with an infinitely large state-action space, they leverage a double layer function approximation routine for dual optimization approximation and robust Q-function estimation. Their proposed algorithm, Robust Fitted Q-Iteration, can scale in applications with continuous states and actions, like CartPole and Hopper. [2] study function approximation in $(s,a)$-rectangular DRMDPs from a theoretical perspective. They present suboptimality bounds depending on the complexity measure of the function class and independent on the size of the state-action spaces.
>
>
> ---
> > **Q4:** how difficult to obtain good estimates of the latent modes.
>
> **A4:** Thanks for the interesting question. To answer this question, we would like to lay out several practical motivations for our framework. Mixture models, in particular Gaussian Mixture Models (GMMs), are widely studied in fields of optimal control and RL. For example:
>
> **Model Predictive Control (MPC):** [1] studies the density steering for discrete-time linear systems, a variant of optimal mass transport problem widely used in applications such as controlling a swarm of robots/drones to achieve a desired spatial distribution. They model the stochastic dynamical system using GMMs and derive its optimal control policy. [2] studies a stochastic MPC algorithm for linear systems subject to additive Gaussian mixture disturbances. They consider a vehicle control case study in which the vehicle must maintain its position on an ill-maintained road.
>
> **Meta RL:** Meta learning is one way to increase the data efficiency of learning algorithms by generalizing learned concepts from a set of training tasks to unseen, but related, tasks. Recent works on Meta RL model the dynamics of a new system as a mixture of expert systems, thus realizing knowledge transfer. In particular, [3,4] model the dynamics of the expert systems using Gaussian process with latent embedding. And the dynamics of the new system is constructed by a mixture of Gaussian processes with the distribution over the latent embedding being the mixture weights, which should be learned from data. In experiments, their meta-learning models effectively generalizes to novel tasks under various environments such as Cart-pole & Double-pendulum swing-up, HalfCheetah and Highway-Intersection.
>
> **World Models for model-based RL:** Model-based Deep Reinforcement Learning (RL) assumes the availability of a model of an environment’s underlying transition dynamics, which are stochastic in nature and oftentimes multimodal. [5,6] use Mixture-density Networks to construct World Models to capture the multimodality in dynamics. Then model-based RL methods are implemented based on the constructed World Models to solve various tasks such as car racing and Inverse Sine Wave.
>
>
> These applications naturally give rise to distributionally robust MDP problems, for example 1) robust control of drones or vehicles is needed to hedge against environmental uncertainties and distributional perturbations; 2) given established expert systems, usually the mixture weights are estimated from limited data collected from the unseen new task, and thus of high uncertainty. Our framework could potentially enhance robustness of decision making in unseen tasks by modeling the weighting parameter uncertainty; 3) by modeling the uncertainty in the World Model (specifically, in the weights), our framework enables robust decision making in model-based RL.
> The basis modes in our work can be the Gaussian components in GMMs, dynamics of expert systems, or mixture components in the Mixture-density Networks. As for the basis modes construction, we can adopt the estimation procedure in prior works. The difficulty should depend on specific tasks. It serves an interesting future research direction to also incorporate the uncertainty in basis modes into consideration for robust decision making.
>
>
> ---
> > **Q5:** Potential limitations of the choice to use mixture models.
>
> **A5:**
> Our model assumes the dynamics is a mixture distribution with finite modes/component distributions. This restricts its application in scenarios where mixture distributions cannot approximate the dynamics well. Furthermore, the current manuscript assumes perfect knowledge of the basis modes and focuses solely on uncertainty in the weighting parameters. However, in practice, these basis modes often need to be estimated from data, introducing an additional layer of uncertainty that can significantly affect decision-making. Addressing this source of uncertainty is nontrivial and falls outside the scope of the current work; we leave it for future investigation.
>
> ---
> **References:**
>
> [1] Shi et. al., 2024, Distributionally robust model-based offline reinforcement learning with near-optimal sample complexity
>
> [2] Blanchet et. al., 2023, Double Pessimism is Provably Efficient for Distributionally Robust Offline Reinforcement Learning: Generic Algorithm and Robust Partial Coverage
>
> [3] Shi et. al., 2023, The Curious Price of Distributional Robustness in Reinforcement Learning with a Generative Model
>
> [4] Wang et. al., 2024, A Unified Principle of Pessimism for Offline Reinforcement Learning under Model Mismatch
>
> [5] Panaganti et. al., 2022, Robust reinforcement learning using offline data
>
> [6] Liu et. al., 2022, Distributionally Robust Q-Learning
>
> [7] Balci et. al., 2024, Density steering of gaussian mixture models for discrete-time linear systems
>
> [8] Engelaar et. al., 2024, Stochastic MPC for Finite Gaussian Mixture Disturbances with Guarantees
>
> [9] Sæmundsson et. al., 2018, Meta reinforcement learning with latent variable gaussian processes
>
> [10] Xu et. al., 2020, Task-agnostic online reinforcement learning with an infinite mixture of gaussian processes
>
> [11] Sedlmeier et. al., 2021, Quantifying multimodality in world models
>
> [12] Ha et. al., 2018, Recurrent world models facilitate policy evolution

---

> > ### Comment · Reviewer_sAJt · 2025-08-04
> > **Response**
> >
> > I thank the authors for their detailed responses to my questions. I will maintain my score.

---

### Official Review · Reviewer_i9w6 · 2025-07-02

**Clarity:** 3
**Significance:** 2
**Originality:** 2
**Rating:** 4
**Confidence:** 4

**Summary:**

This paper introduces a framework for distributionally robust reinforcement learning, namely the linear mixture DRMDP, where the uncertainty in dynamics is modeled as perturbations to the mixture weights $\theta$ in a known set of basis transition kernels $\phi$. The authors propose a meta-algorithm that first estimates the nominal model through regression, then applies a doubly pessimistic policy optimization procedure. They analyze the sample complexity of this approach under total variation, KL, and $\chi^2$ divergence-based uncertainty sets, providing suboptimality bounds in the offline RL setting.

**Questions:**

Are there any practical examples that naturally fit the linear mixture structure assumed in this paper? In many real-world applications where model dynamics are uncertain and robustness is desired, it often seems more appropriate to model uncertainty in the features
$\phi$, rather than in the mixture weights $\theta$.

**Ethical Concerns:**

["NO or VERY MINOR ethics concerns only"]

**Final Justification:**

The rebuttal clarifies my major doubts and concerns. However, I still think that the practical contribution seems limited and the authors makes no efforts in terms of validating there approach even on some simple practical examples.

**Limitations:**

Yes

**Paper Formatting Concerns:**

None.

**Quality:**

2

**Strengths And Weaknesses:**

The paper is well-organized and clearly written, which makes it easy to follow the technical developments. Robustness in reinforcement learning—especially in the offline and off-dynamics setting—is a meaningful and timely topic, and the proposed linear mixture DRMDP framework offers an interesting perspective that incorporates structural assumptions into the uncertainty set design. Although I have not verified all the proofs in detail, the analysis appears technically sound.

A key concern I have is regarding the practical motivation for the linear mixture distributionally robust setting. While the theoretical framework is cleanly laid out, the paper does not provide any concrete or compelling real-world example where such a structure naturally arises. The simulated environments used in Appendix F appear toy-like, which makes it difficult to judge whether the linear mixture structure meaningfully captures uncertainty in practical applications, or if it was primarily chosen for analytical convenience.

Additionally, the algorithm itself appears relatively straightforward: first estimate a nominal model via regression, then apply a "double pessimism" policy optimization using that estimate. The learning and planning phases are essentially decoupled, and the optimization uses known techniques adapted to this specific uncertainty set. While the theoretical guarantees are solid, the technical depth of the algorithmic development seems limited, and the analysis benefits from the structure already assumed.

---

> ### Author Rebuttal · Authors · 2025-07-31
>
> We thank the reviewer for their valuable time and effort in providing detailed feedback on our work. We hope our response will fully address all of the reviewer's points.
>
> ---
> > **Q1:** Practical motivation for the linear mixture distributionally robust setting.
>
> **A1:** Mixture models, in particular Gaussian Mixture Models (GMMs), are widely studied in fields of optimal control and RL. For example:
>
> **Model Predictive Control (MPC):** [1] studies the density steering for discrete-time linear systems, a variant of optimal mass transport problem widely used in applications such as controlling a swarm of robots/drones to achieve a desired spatial distribution. They model the stochastic dynamical system using GMMs and derive its optimal control policy. [2] studies a stochastic MPC algorithm for linear systems subject to additive Gaussian mixture disturbances. They consider a vehicle control case study in which the vehicle must maintain its position on an ill-maintained road.
>
> **Meta RL:** Meta learning is one way to increase the data efficiency of learning algorithms by generalizing learned concepts from a set of training tasks to unseen, but related, tasks. Recent works on Meta RL model the dynamics of a new system as a mixture of expert systems, thus realizing knowledge transfer. In particular, [3,4] model the dynamics of the expert systems using Gaussian process with latent embedding. And the dynamics of the new system is constructed by a mixture of Gaussian processes with the distribution over the latent embedding being the mixture weights, which should be learned from data. In experiments, their meta-learning models effectively generalizes to novel tasks under various environments such as Cart-pole & Double-pendulum swing-up, HalfCheetah and Highway-Intersection.
>
> **World Models for model-based RL:** Model-based Deep Reinforcement Learning (RL) assumes the availability of a model of an environment’s underlying transition dynamics, which are stochastic in nature and oftentimes multimodal. [5,6] use Mixture-density Networks to construct World Models to capture the multimodality in dynamics. Then model-based RL methods are implemented based on the constructed World Models to solve various tasks such as car racing and Inverse Sine Wave.
>
>
> The basis modes in our work can be the Gaussian components in GMMs, dynamics of expert systems, or mixture components in the Mixture-density Networks. These applications naturally give rise to distributionally robust MDP problems, for example 1) robust control of drones or vehicles is needed to hedge against environmental uncertainties and distributional perturbations; 2) given established expert systems, usually the mixture weights are estimated from limited data collected from the unseen new task, and thus of high uncertainty. Our framework could potentially enhance robustness of decision making in unseen tasks by modeling the weighting parameter uncertainty; 3) by modeling the uncertainty in the World Model (specifically, in the weights), our framework enables robust decision making in model-based RL.
>
>
> ---
> > **Q2:** In many real-world applications where model dynamics are uncertain and robustness is desired, it often seems more appropriate to model uncertainty in the features $\phi$, rather than in the mixture weights $\theta$.
>
> **A2:** Thank you for raising this insightful point. Take the Meta-RL application in our response to Q1 for an example, the expert systems are constructed from huge amounts of historical data and thus of low uncertainty. Once constructed, they can be deemed as fixed. To estimate the dynamics of the new tasks, we have to estimate the mixture weights from data collected from the new tasks, which is usually of limit size and cause large uncertainty in the mixture weights estimation. This motivates us to model the uncertainty in the mixture weights $\theta$ to enable robust decision making in the new task, which turns out to be natural and of great importance.
>
> Moreover, we agree that accounting for the uncertainty in the feature $\phi$ is also important for achieving robust decision-making. However, addressing it requires careful and non-trivial treatment in both algorithm design and theoretical analysis—particularly due to its interaction with uncertainty in the weighting parameters. A thorough investigation of this issue is beyond the scope of our current work, and we leave it as an important direction for future research.
>
> ---
> > **Q3:** Algorithm design: Decoupling of the learning and planning phase
>
> **A3:** Thank you for this interesting question. We would like to provide more insights on our choice of algorithm design. We highlight that the decoupling of the learning and planning phase is the key to make theoretical guarantees possible. In particular, the analysis of the meta-algorithm only needs to control one-source of estimation error, which arises from the transition-targeted estimation of $\theta_h$. Moreover, the transition-targeted estimation simply uses $\phi$ as the feature vector. These two attributes are essential to allow Assumptions 3.4, 3.10 and 3.14 to neatly control the coverage of the offline dataset. And our analysis also leverages recently established concentration lemma for dependent error structure as we discussed in Remark 4.2.
>
> For algorithms that embed learning with learning step-wisely, like the value-targeted regression algorithm widely studied in standard linear mixture MDP literature [7], they involve two layers of estimation error, i.e., the estimation error in $\theta$ and the estimation error in $\hat{V}$. During the preparation of this work, we found that the consideration of dynamics shift in DRMDP induces essential hardness to control these two sources of estimation error, unlike in standard MDPs. It is still unclear how to quantify/control the data coverage of the offline dataset with this type of algorithm. We would like to leave this problem for future study.
>
>
> ---
> **References:**
>
> [1] Balci et. al., 2024, Density steering of gaussian mixture models for discrete-time linear systems
>
> [2] Engelaar et. al., 2024, Stochastic MPC for Finite Gaussian Mixture Disturbances with Guarantees
>
> [3] Sæmundsson et. al., 2018, Meta reinforcement learning with latent variable gaussian processes
>
> [4] Xu et. al., 2020, Task-agnostic online reinforcement learning with an infinite mixture of gaussian processes
>
> [5] Sedlmeier et. al., 2021, Quantifying multimodality in world models
>
> [6] Ha et. al., 2018, Recurrent world models facilitate policy evolution
>
> [7] Jia et. al., 2020, Model-based reinforcement learning with value-targeted regression.
>
>
> ---
> We hope we have addressed all of your questions/concerns. If you have any further questions, we would be more than happy to answer them and if you don’t, would you kindly consider increasing your score?

---

> > ### Comment · Reviewer_i9w6 · 2025-08-05
> >
> > I would like to thank the authors for the thorough rebuttal that clears most of my concerns. I will raise my score.

---

### Official Review · Reviewer_8wNc · 2025-07-03

**Clarity:** 2
**Significance:** 3
**Originality:** 3
**Rating:** 4
**Confidence:** 4

**Summary:**

This paper introduces a novel framework called linear mixture DRMDPs to tackle the off-dynamics reinforcement learning problem, where an agent must learn a policy in a source environment but deploy it in a target environment with different dynamics. This work assumes that both the source and target transition kernels share a linear mixture structure and proposes a meta-algorithm for offline robust policy learning in this setting using transition-targeted ridge regression and a double pessimism principle. The authors derive finite-sample guarantees for learning an approximately optimal robust policy under three different f-divergence-based uncertainty sets.

**Questions:**

I have a few follow-up questions that I understand may be beyond the intended scope of this work. However, I would greatly appreciate it if the authors could provide any insights or clarifications in their response:
1. How sensitive is the framework to mis-specified basis modes in practice?
2. Would it be possible to extend the framework to linear MDPs and low-rank MDPs?
3. Would it be possible to extend the framework to online settings with exploration guarantees?

**Ethical Concerns:**

["NO or VERY MINOR ethics concerns only"]

**Final Justification:**

I appreciate the authors' detailed responses to my questions. I will maintain my rating.

**Limitations:**

yes

**Quality:**

3

**Strengths And Weaknesses:**

**Strengths:**

1. The paper introduces a new variant of DRMDPs that leverages linear mixture structures in the transition dynamics. This is a compelling and practically motivated idea that brings additional structure to the uncertainty set, going beyond conventional (s,a)- or d-rectangular formulations.
2. The proposed framework is rigorously analyzed under three widely used f-divergences. The authors derive finite-sample suboptimality bounds, showing the statistical learnability of robust policies under the linear mixture setting.
3. The linear mixture uncertainty sets are shown, both theoretically and illustratively, to be strictly smaller than standard rectangular sets in several scenarios. This leads to less conservative policies and tighter robustness guarantees.
4. The proposed algorithm is modular and adaptable, capable of working with different divergence metrics and scalable with known basis modes. The adoption of transition-targeted regression is also well-motivated given the limitations of value-targeted methods in this context.

**Weaknesses:**

1. The meta-algorithm depends on a planning oracle that optimizes over a doubly pessimistic objective involving both parameter and transition uncertainty. As acknowledged by the authors, this oracle is computationally intractable, and no approximation strategy is integrated into the theoretical analysis. This creates a gap between theory and practice.
2. While the specific combination of linear mixture dynamics and distributional robustness is new, many of the underlying components—such as f-divergence uncertainty sets, ridge regression for parameter estimation, and pessimistic value estimation—have been studied extensively. The novelty lies in integrating these ideas, but the conceptual leap is incremental compared to prior work on d-rectangular or linear DRMDPs.
3. Some theoretical sections (e.g., the coverage assumptions and concentration analysis in Section 4) are dense and could benefit from clearer intuition or visual aids. This may hinder accessibility to readers unfamiliar with the robust RL literature.

---

> ### Author Rebuttal · Authors · 2025-07-31
>
> We thank the reviewer for their valuable time and effort in providing detailed feedback on our work. We hope our response will fully address all of the reviewer's points.
>
> ---
> > **Q1:**  The theoretical section is dense and could benefit from clearer intuition or visual aids.
>
>
> **A1:** Thank you for the suggestion. The current layout of section 4 is due to the space limit. Given additional space in the camera ready, we will separate section 4 into two sections, Algorithm Design and Theoretical Guarantees, and expand our current discussion and interpretation on our analysis and assumptions.
>
>
> ---
> > **Q2:** How sensitive is the framework to mis-specified basis modes in practice?
>
>
> **A2:** From a theoretical point of view, mis-specification in basis modes can cause error and result in suboptimal performance, then there would be no guarantee for the algorithm to learn the optimal robust policy. In practice, the basis modes are typically constructed from data, such as the Gaussian components in Gaussian Mixture Model [1,2], or dynamics of expert systems in Meta-RL [3,4], thus approximation error is inevitable. Nevertheless, these prior works have shown certain empirical success in using mixture models to approximate the complex true dynamics and performances actually vary from tasks to tasks. Another empirical evidence is from the recent work [5] on d-rectangular linear DRMDPs. They use linear MDP to model nonlinear dynamics. Even with large mis-specification errors in the feature mapping, their algorithm still achieves certain robustness performance. Based on these evidences, we conjecture that sensitivity of our framework with respect to the misspecification in basis modes should depend on specific tasks. But even with certain approximation errors, we can still expect certain robustness performances in practice. It serves an interesting future research direction to include the uncertainty in basis modes into consideration.
>
>
> ---
> > **Q3:** Extend the framework to linear MDPs and low-rank MDPs.
>
> **A3:** We would like to note that distributionally robust MDPs have been separately studied in prior works under the framework of linear MDPs [5] and low rank MDPs [6]. They study different types of uncertainty sets constructed based on the posed structures of transition models, propose corresponding algorithms and provide theoretical guarantees. In our work, we particularly focus on a mixture structure of the dynamics and modeling uncertainty in the mixture weights. Thus, our work is orthogonal to previous ones on linear MDPs and low-rank MDPs.
>
>
> ---
> > **Q4:** Would it be possible to extend the framework to online settings with exploration guarantees?
>
> **A4:** Thanks for the great question. Yes, our framework can potentially be extended to online settings where the agent has to actively interact with the nominal environment to collect data. Challenges lie in how to design algorithms capable of reasoning about the exploration and address the exploration-exploitation trade off in a data efficient manner. A potential solution would be leveraging the principle of optimism in the face of uncertainty [7]. For example, at each episode $k$, we can maintain a confidence set $\hat{\mathcal{P}}^k$ based on the weighting parameter $\theta$ estimation and update it when new trajectories are collected. Then we define the value estimator $J^k(\pi)$ as the largest robust value function with nominal kernel included in the confidence set: $J^k(\pi) = \sup_{P\in \hat{\mathcal{P}}^k}\inf_{\tilde{P}\in \mathcal{U}_h^{\rho}(P)}V_{\tilde{P}}^{\pi}(s_1)$, instead of the double pessimism estimator in our work. Then we use the optimal policy wrt $J^k(\pi)$ as the execution policy at the current episode to collect data. Another potential challenge would be quantifying the difficulty of exploration in online DRMDPs. There is a line of work studying this problem and several interesting phenomenons have been identified that are unique to the online setting. We leave a rigorous theoretical development for future research.
>
> ---
> **References:**
>
> [1] Balci et. al., 2024, Density steering of gaussian mixture models for discrete-time linear systems
>
> [2] Engelaar et. al., 2024, Stochastic MPC for Finite Gaussian Mixture Disturbances with Guarantees
>
> [3] Sæmundsson et. al., 2018, Meta reinforcement learning with latent variable gaussian processes
>
> [4] Xu et. al., 2020, Task-agnostic online reinforcement learning with an infinite mixture of gaussian processes
>
> [5] Liu et. al., 2024, Distributionally robust off-dynamics reinforcement learning: Provable efficiency with linear function approximation
>
> [6] Hu et. al., 2024, Efficient Duple Perturbation Robustness in Low-rank MDPs
>
> [7] Azar et. al., 2017, Minimax Regret Bounds for Reinforcement Learning
>
> [8] He et. al., 2025, Sample Complexity of Distributionally Robust Off-Dynamics Reinforcement Learning with Online Interaction
>
> ---
> We hope we have addressed all of your questions/concerns. If you have any further questions, we would be more than happy to answer them and if you don’t, would you kindly consider increasing your score?

---

> > ### Comment · Reviewer_8wNc · 2025-08-04
> >
> > Thank you for the detailed responses to my questions. I will maintain my score.

---

### Decision · Program_Chairs · 2025-09-17

**Decision:**

Accept (poster)

**Comment:**

Summary: This paper introduces a new class of distributionally robust Markov decision processes (DRMDPs) that leverage linear mixture structures in the transition dynamics. Unlike conventional rectangular uncertainty sets, the proposed linear mixture ambiguity sets exploit structural assumptions to yield less conservative policies and tighter robustness guarantees. The authors develop a modular algorithm, provide rigorous theoretical analysis under several f-divergences, and establish finite-sample suboptimality bounds that demonstrate the statistical learnability of robust policies. Illustrative examples highlight how the linear mixture sets can be strictly smaller than standard ones.

Comments: We received 4 expert reviews, with scores 4, 4, 4, 5, and the average score is 4.25.

The reviewers acknowledge that the paper is technically solid, with an interesting theoretical contribution in robust RL. In particular, the linear mixture ambiguity sets are shown to provide advantages over standard (s,a)- and d-rectangular sets, both theoretically and illustratively. The work establishes meaningful finite-sample guarantees and variance bounds. The paper is well-written.

The reviewers have also pointed out many weaknesses. Reviewer 8wNc has pointed out that the planning oracle assumed in the theoretical framework is computationally intractable, and no approximation method is analyzed, creating a gap between theory and practical deployment. More than one reviewer has pointed out that the experimental validation is minimal.

Based on my reading of the paper, reviews, and rebuttal discussion, I believe that, overall, this paper provides strong theoretical contributions and advances the literature on robust RL by proposing and analyzing a new ambiguity set structure. While the lack of empirical evaluation is a limitation, the primary strength lies in the theory, and the technical results are substantial enough to merit acceptance. The authors should, however, improve the presentation of the intuition behind their assumptions, clarify the conditions under which linear mixture ambiguity sets offer clear advantages over existing alternatives, and provide a more explicit discussion of potential applications. Including even modest additional experiments or comparisons to standard rectangular sets would significantly strengthen the case for practical impact.